# CHIMERE-2017: From urban to hemispheric chemistry-transport modeling

**Sylvain Mailler**[1,2], **Laurent Menut**[1], **Dmitry Khvorostyanov**[1], **Myrto Valari**[1], **Florian Couvidat**[3], **Guillaume Siour**[4], **Solène Turquety**[1], **Régis Briant**[1], **Paolo Tuccella**[1], **Bertrand Bessagnet**[3], **Augustin Colette**[3], **Laurent Létinois**[3], **Kostantinos Markakis**[1], **and Frédérik Meleux**[3]

[1]LMD/IPSL, École Polytechnique, Université Paris Saclay, ENS, PSL Research University; Sorbonne Universités, UPMC Univ Paris 06, CNRS, Palaiseau, France
[2]École des Ponts ParisTech, Université Paris-Est, 77455 Champs-sur-Marne, France
[3]INERIS, National Institute for Industrial Environment and Risks, Parc Technologique ALATA, F-60550 Verneuil-en-Halatte, France
[4]Laboratoire Interuniversitaire des Systèmes Atmosphériques (LISA), UMR CNRS 7583, Université Paris Est Créteil et Université Paris Diderot, Institut Pierre Simon Laplace, Créteil, France

*Correspondence to:* Sylvain Mailler (sylvain.mailler@lmd.polytechnique.fr)

**Abstract.** CHIMERE is a chemistry-transport model designed for regional atmospheric composition. It can be used at a variety of scales from local to continental domains. However, due to the model design and its historical use as a regional model, major limitations had remained, hampering its use at hemispheric scale, due to the coordinate system used for transport as well as to missing processes that are important in regions outside Europe. Most of these limitations have been removed in the CHIMERE-2017 version, allowing its use in any region of the world and at any scale, from the scale of a single urban area up to hemispheric scale, with or without polar regions included. Other important improvements have been made in the treatment of the physical processes affecting aerosols and the emissions of mineral dust. From a computational point of view, the parallelization strategy of the model has also been updated in order to improve model numerical performance and reduce the code complexity. The present article describes all these changes. Statistical scores for a model simulation over continental Europe are presented, and a simulation of the circumpolar transport of volcanic ash plume from the Puyehue volcanic eruption in June 2011 in Chile provides a test case for the new model version at hemispheric scale.

## 1 Introduction

Deterministic chemistry-transport modelling is now widely used for the analysis of pollution events, scenarios and forecast (Monks et al., 2009). Numerous models exist and are used from local to global scale, both for gaseous and aerosols modelling (Simpson et al. (2012); Inness et al. (2013) among many others). While models were previously dedicated mainly to specific processes, the latest generation of chemistry-transport models (CTMs) aims at representing the complete set of processes leading to changes in the atmospheric composition in terms of aerosols and trace gases. For regional air quality in the troposphere, several CTMs are currently developed and are able to include all types of emissions: anthropogenic, biogenic, mineral dust, sea salt, vegetation fires and volcanos. Even though all these emission processes are now included in many CTMs, the emitted species have different chemistry and lifetimes, and models often address some specific applications and thus specific spatial areas. This was the case of the CHIMERE model, extensively described in Menut et al. (2013a) for its 2013 version. Originally, CHIMERE was designed for urban areas. It was extended later to western Europe, and then to the northern part of Africa by including mineral dust emissions, but was limited to these areas only, due to limitations in available data (such as the anthropogenic emissions). The typical resolution (grid-spacing) of the simulation domains range from 4 km

for urban-scale domains to about 50 km for regional-scale domains Markakis et al. (2015); Valari and Menut (2008).

The CHIMERE model has been used for a long time for studies at the urban to regional scale. Vautard et al. (2007) has used this model within the CityDelta project over four major urban areas in Europe (Berlin, Milan, Paris and Prague), at a horizontal resolution of 5 km. While this resolution is not sufficient to resolve adequately urban-scale phenomena, Valari and Menut (2008) have shown that due to limitations in the accuracy of the input meteorological fields, increasing the horizontal model resolution to values lower than 10 km might actually degrade model performance. The same authors (Valari and Menut, 2010), show that, actually, rather than increasing the model resolution towards kilometric scale, better results can be obtained by downscaling model results to a kilometric resolution representative of urban scale by mixing model outputs with fine scale information on emissions. Recent studies using CHIMERE at urban scale include the work of Markakis et al. (2015), using a set of long-term (10 year) CHIMERE simulations at 4km horizontal resolution for the Paris region, including urban, suburban and rural areas, where the CHIMERE model is used for the present climate but also to test the possible impact of different emission and climate scenarios on air quality in this area. CHIMERE has also been used at continental scale for a long time, including model intercomparison exercises such as AQMEII (Rao et al., 2011; Solazzo et al., 2012b, a), Eurodelta (Schaap et al., 2007) and more recently Eurodelta III Bessagnet et al. (2016). The latter study presents the evaluation of the CHIMERE outputs for the main species of gaseous and particulate atmospheric trace components along with these of six other state-of-the-art models over Europe. The interested reader is therefore referred to Bessagnet et al. (2016) for a detailed comparison of the CHIMERE characteristics and performance compared to other models, and to Terrenoire et al. (2015) for a detailed overview of the CHIMERE performance and scores regarding the concentrations of many gaseous and aerosol species compared to a network of ground measurements over Europe for year 2009). As these studies at continental scale are very recent and dramatic changes in model performance over Europe do not occur from the changes presented here, the present article is not only focused on evaluating the model performance relative to observations but rather on describing the generalization of the model scope to hemispheric scales and the inclusion of new processes. For forecasts, the model is applied daily for the French PREVAIR system, (Honoré et al., 2008), the COPERNICUS program, (Copernicus, 2017), as well as in many air quality networks.

In this paper, the CHIMERE-2017 model version is presented. All new developments made since the CHIMERE-2013 version (Menut et al., 2013a) are presented. This mainly consists in an extension of input databases, model grid management, optimization and chemical mechanism. The changes for the grid management are dedicated to build a

CTM able to run over a hemispheric domains as well as for smaller regions anywhere in the world. These developments required important changes in the model, as well as the improvement of many processes already included in the previous version: the Fast-JX module for realistic evaluation of the photolysis rates has been added and allows the calculation of updated photolysis rates at each physical time step, including the optical effects of clouds and aerosols. The mineral dust emissions have been upgraded in order to estimate fluxes in any region. In addition, this new version has also been an opportunity to update the representation of chemical processes by giving the user the choice to use the SAPRC chemical mechanism, which is more widely used than the MELCHIOR chemical scheme developed for the CHIMERE model (Lattuati, 1997; Menut et al., 2013a). Chlorine chemistry has been included, and the representation of physical processes affecting the aerosols, such as nucleation, coagulation, and wet deposition has been improved, while a scheme for traffic-related resuspension of particulate matter in urbanized areas has been included in the model.

CHIMERE-2017 is an offline chemistry-transport model, meaning that it needs to be provided with input meteorological fields, and does not implement any feedback of atmospheric chemistry on atmospheric dynamics. As the CHIMERE model is used for both analysis and forecast, particular attention was given to the optimization of computational performance. Numerous improvements were made in the code and are completely transparent for the user: these changes are described in Section 2.

Section 3 presents the changes in the model geometry, including the vertical mesh, as well as changes in the horizontal coordinate system allowing the application of the model to hemispheric scale domains.

Section 4 presents the improvements in the representation of anthropogenic emissions, including the use of the global HTAP emission dataset for anthropogenic emissions, and the improvements in modelling mineral dust emissions.

Section 5 describes the changes in the representation of various physical and chemical processes in the model, such as inclusion of the SAPRC scheme for gaseous chemistry and inclusion of chlorine chemistry in the model. This section also presents the evolutions in the modelling of the physical processes affecting aerosols, as well as the implementation of the Fast-JX module for radiative transfers. Another major improvement presented in this section is the ability of CHIMERE-2017 to provide LIDAR observables as a model output.

Section 6 presents the application of CHIMERE-2017 to simulations of three winter months and three summer months in a domain covering continental Europe at 50 km resolution, and the scores obtained by the model in comparison with background observations of gaseous and particulate species in this configuration.

Section 7 presents the application of the new model version to the simulation of the eruption of the Puyehue-Cordon

Caulle volcano, in the Chilean Andes, in June 2011. This event provides a good testbed for this new version, since the volcanic plume from this volcanic eruption was dense enough to be observed by satellites all along its circumpolar transport around the South Pole.

Finally, Section 8 presents the conclusions of the present study, in terms of applications made possible by this new model version, as well as the outlines for future developments of the CHIMERE model.

## 2 Optimizations

Several technical changes were made in the CHIMERE code to improve code scalability: these changes regard the parallelization of many preprocessors into the parallelized section of the model, along with improvement of the parallelization strategy for some parts of the model that were already parallelized in order to improve code scalability.

### 2.1 Parallelization of preprocessors

Compared to the previous model version, several programs that used to be sequential preprocessors executed before the CHIMERE run itself have now been parallelized and included into the main CHIMERE executable. This is the case of the interpolation and treatment of the input meteorological fields. In the new model version, these fields are read and processed at each hourly time step (instead of being processed once and for all in a sequential way at the beginning of the run). This new design has no impact on the model outputs but has two advantages:

1. It allows a reduction of computation time by parallelization of this calculation step

2. It enables the possibility to develop an online coupled version of the model, in which case the meteorological fields would not be pregenerated.

Note that this "real-time" processing of the meteorological fields is only available for users who use meteorological fields from WRF. For users of other sources of meteorological data such as ECMWF products, offline meteorological preprocessors are still provided with the model. Another important point is that even though the processing of meteorological input has been changed as described here, the version presented here does not take into account any radiative or microphysical feedback of atmospheric chemistry on meteorology. A version including aerosol-radiation interactions through online coupling of CHIMERE with WRF has been developed (Briant et al., 2017), and is available upon request from the lead author of that study. Apart from allowing online coupling between CHIMERE and WRF, the model setup described by Briant et al. (2017) also permits to update the meteorological fields at any timestep shorter than one hour.

Table 1 lists the variables that can be read by CHIMERE from the outputs of the meteorological model, separating the variables that are mandatory from the optional ones.

### 2.2 Improvement of the parallelization

In 2006, the main CHIMERE loop was parallelized using a master/slave pattern. A cartesian division of the simulation domain into several sub-domains is done, each sub-domain being attributed to one slave process. Each slave performs the model integration in its own geographical sub-domain as well as boundary condition exchanges with its neighbours in order to permit transport from one slave to the next. In addition, in former CHIMERE versions, a master process was needed in order to gather and scatter data from the various slave processes that performed the actual gridded calculations, and to perform initializations and file input/output.

The use of a master process limited the efficiency of the parallelized code, since the master process did not perform any computation except gathering and scattering the data to and from the slaves, and that it totally centralized the input and output tasks, a bottleneck effect which limited the gains realized by parallelization, particularly when the simulation domains were very large and split between many slaves.

Therefore, in the CHIMERE-2017 version, this master process has been removed: using the parallel input/output routines of the Parallel-Netcdf library (Li et al., 2003), each slave process now reads the netcdf input files and writes the output data for its own sub-domain into a single output netcdf file common to all slaves, removing the bottleneck effect due to the centralization of input/output tasks.

This induces some major simplifications of CHIMERE code, including reduction of inter-process communications related to the parallelization of the input/output processes, which were performed in a central way by the master process in previous model version.

## 3 Model geometry

Major changes have been implemented in CHIMERE-2017 compared to earlier CHIMERE versions, opening the possibility to perform simulations in domains including the pole.

Historically, CHIMERE was first designed as a box model for the region of Paris (Menut et al., 2000). Rapidly, it has been transformed into a cartesian model on curvilinear Arakawa C-grids (Arakawa and Lamb (1977), see Fig. 1). However, the formulation of the transport scheme on these curvilinear grids up to CHIMERE-2014b was still based on a lat-lon formulation, which implied the impossibility to include poles in the domain. In CHIMERE-2017, as in earlier versions, the user can choose between three different options for horizontal transport schemes, namely the basic upwind scheme, the slope-limited Van Leer scheme (Van Leer, 1979), and the Piecewise-parabolic

| CHIMERE name | Variable | Dimension | Units |
|---|---|---|---|
| **Mandatory Variables** | | | |
| lon | longitude of gridpoints | 2D | degrees_east |
| lat | latitude of gridpoints | 2D | degrees_north |
| tem2 | 2m Temperature | 2D | K |
| soim | Soil moisture | 2D | $m^3/m^3$ |
| rh2m | 2m Relative humidity | 2D | 0-1 |
| lspc | Large-scale Precipitation | 2D | $kg/m^2/hour$ |
| copc | Convective Precipitation | 2D | $kg/m^2/hour$ |
| temp | Temperature | 3D | K |
| cliq | Cloud liquid water content (excluding rain water) | 3D | Kg/Kg |
| sphu | Specific humidity | 3D | kg/kg |
| pres | Pressure | 3D | Pa |
| alti | Altitude of half layer | 3D | m |
| winz | Zonal component of the wind | 3D | m/s |
| winm | Meridional component of the wind | 3D | m/s |
| swrd | Short Wave Radiation | 2D | $W/m^2$ |
| **Optional Variables** | | | |
| lwrd | Long Wave Radiation | 2D | $W/m^2$ |
| sshf | Surface sensible heat flux | 2D | $W/m^2$ |
| slhf | Surface latent heat flux | 2D | $W/m^2$ |
| usta | Friction velocity | 2D | m/s |
| hght | Boundary layer height | 2D | m |
| weas | Water equivalent accumulate snow | 2D | $kg/m^2$ |
| snowh | Snow height | 2D | m |
| seaice | Sea-ice ratio | 2D | *n/a* |
| psfc | Surface pressure | 2D | Pa |
| rain | rain water content | 3D | Kg/Kg |
| cice | ice content | 3D | Kg/Kg |

**Table 1.** *Mandatory and optional variables obtained from meteorological input data. If the optional variables are not provided by the raw meteorological model, there are diagnosed during the simulation.*

method (Colella and Woodward, 1984), all of which are examined in the CHIMERE model in Vuolo et al. (2009). These three schemes are designed to estimate the trace species concentration at grid cell interfaces in order to convert the mass flux of total air through cell boundaries into mass fluxes for each of the model species through these boundaries. While the implementation of these schemes has needed no change in building the present model version, the estimate of the atmospheric mass flux between neighbouring model grid cells has been revised by switching to a new coordinate system in order to lift model limitations concerning the geographic poles and the date-change lines. These three schemes are designed to be monotonous (because they include the use of slope-limiting algorithms, except the Upwind scheme which does not need the use of such algorithm), and mass-conservative because of their flux formulation.

This has been achieved by switching from a representation of the grid points in a spherical lat-lon coordinate system, singular at the pole, to a 3d cartesian coordinate system, which has no singularity. In the former CHIMERE versions the grid centers were represented by their geographical coordinates $\left(\lambda^{ij}, \phi^{ij}\right)$, and the wind vectors by their projection on the local frame $(\mathbf{u}_\lambda, \mathbf{u}_\phi)$ (Fig. 2). In the present version, the points are represented by their cartesian coordinates in the frame centered at the Earth center and with unit vectors $(\mathbf{u}_1, \mathbf{u}_2, \mathbf{u}_3)$, and the wind vectors are represented by their projections on these unit vectors.

This change in the internal representation of spherical geometry has only a small impact on the simulated values, in the sense that it corrects some geometrical errors that appeared due to the assumptions made in the old coordinate system, but these differences have been found to be of very small amplitude, except in the vicinity of the pole where distorsions due to the lat-lon system become critical. The new coordinate system allows domains that include the pole, without the need for any particular filtering. This strategy allows the creation of regional domains from local to hemispheric scale anywhere on the globe, including one pole or even, which opens possible application of CHIMERE-2017 for studies in the polar areas, including circumpolar transport of polluted air masses, as will be shown in Section 7. An example grid on which CHIMERE-2017 can be run is shown on Fig. 3. This grid is a polar stereographic grid centered at the north pole, entirely covering the northern hemisphere, and

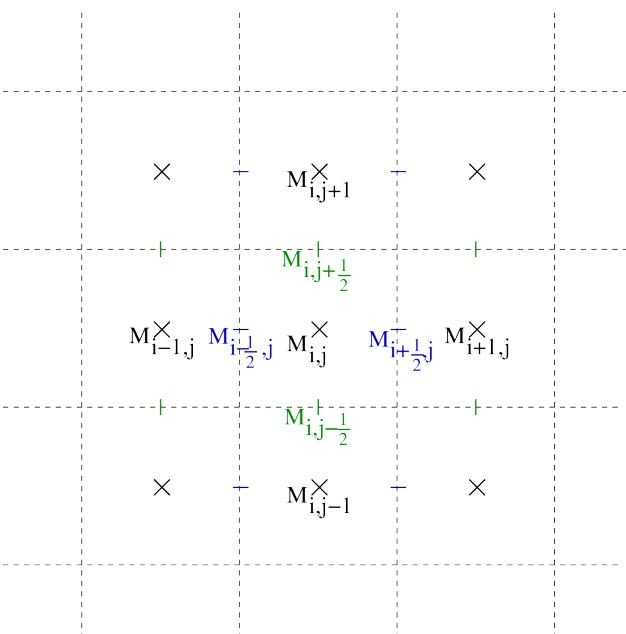

**Figure 1.** *Centered (black) and staggered (blue and green) grid points in the Arakawa C-grid.*

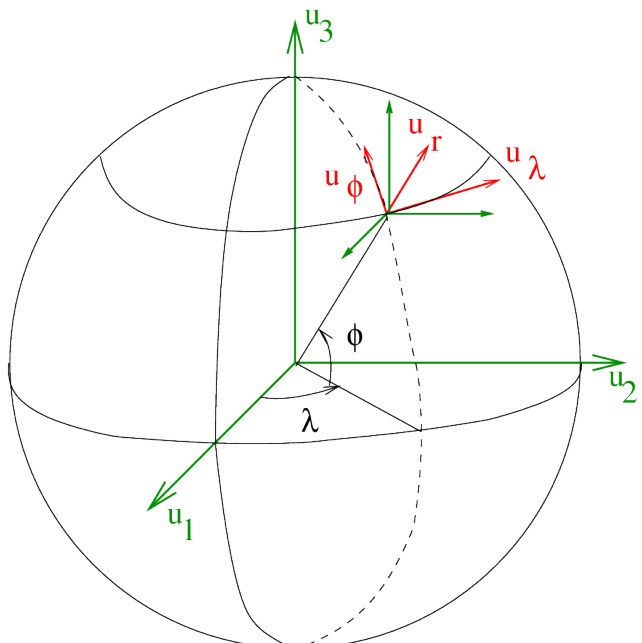

**Figure 2.** *Cartesian and spherical frames for the representation of point coordinates and speed vectors*

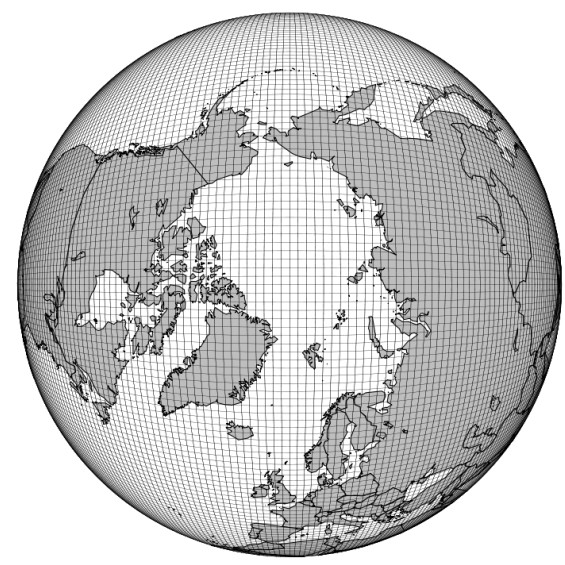

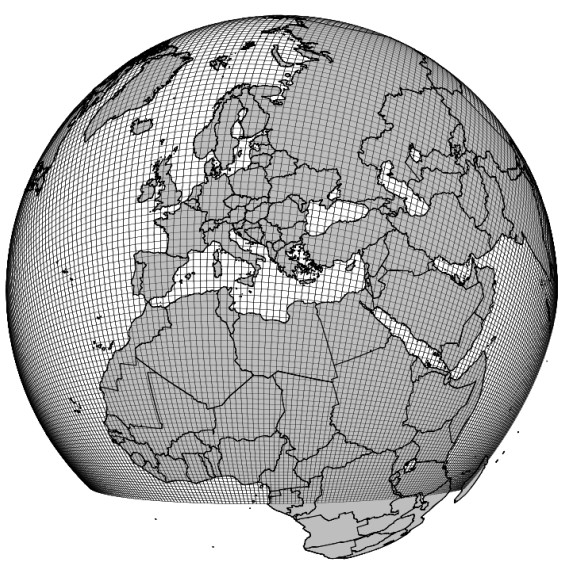

**Figure 3.** *Model grid generated for the northern hemisphere with 180x180 points in polar stereographic projection, viewed from the top (upper panel) and from the side (lower panel).*

with the four corners of the domains extending slightly into the southern hemisphere (as far south as 19.47°S). With this projection and this number of points, the horizontal model resolution varies from 140x140 km$^2$ at the pole to 70x70 km$^2$ at the Equator.

In this new coordinate system, the transport is calculated as follows. First, the coordinates of every grid center $\mathrm{M}^{ij}$ are converted from their geographical coordinates $\left(\lambda^{ij}, \phi^{ij}\right)$

to cartesian coordinates $\left(x_1^{ij}, x_2^{ij}, x_3^{ij}\right)$ on a unit sphere as follows:

$$\begin{cases} x_1^{ij} &= \cos\phi^{ij}\cos\lambda^{ij} \\ x_2^{ij} &= \cos\phi^{ij}\sin\lambda^{ij} \\ x_3^{ij} &= \sin\phi^{ij} \end{cases} \tag{1}$$

The horizontal wind vector $\mathbf{U}^{ij}$ at the grid center is initially represented by the two classical wind components: $\mathbf{U}^{ij} = u^{ij}\cdot\mathbf{u}_\lambda + v^{ij}\cdot\mathbf{u}_\phi$ where the zonal and meridional wind components $u^{ij}$ and $v^{ij}$ are obtained from the meteorological inputs. Since this representation splitting the horizontal wind into a zonal and a meridional component is singular at the geographical poles, before performing the transport operations, the horizontal wind is split into its three components on the cartesian frame $(\mathrm{u}_1, \mathrm{u}_2, \mathrm{u}_3)$ using the following formulae for projecting the wind on the cartesian frame $(\mathbf{u}_1, \mathbf{u}_2, \mathbf{u}_3)$:

$$\begin{cases} U_1^{ij} &= -\sin\lambda u^{ij} - \sin\phi\cos\lambda v^{ij} \\ U_2^{ij} &= \cos\lambda u^{ij} + \sin\phi\sin\lambda v^{ij} \\ U_3^{ij} &= \cos\phi v^{ij} \end{cases} \tag{2}$$

Once the cartesian coordinates of the grid centers $\left(x_1^{ij}, x_2^{ij}, x_3^{ij}\right)$ and of the wind-speed vectors $\left(U_1^{ij}, U_2^{ij}, U_3^{ij}\right)$ are computed at the grid centers, it is easy to obtain the values of the speed vectors at the staggered cells (Fig. 1) with the following formulae:

$$\begin{cases} U_k^{i+\frac{1}{2},j} &= \dfrac{U_k^{ij} + U_k^{i+1,j}}{2} \, (k=1,2,3) \\ U_k^{i,j+\frac{1}{2}} &= \dfrac{U_k^{ij} + U_k^{i,j+1}}{2} \, (k=1,2,3) \end{cases} \tag{3}$$

This new formulation with the use of cartesian coordinates instead of geographical latitude-longitude coordinates for the transport of pollutants removes the constraints that prevented the use of CHIMERE on domains including a geographic pole and/or a date-change line. This new formulation has been tested on the case of the eruption of the Puyehue volcano, in June 2011, a case during which the ash plume from the volcano went around the south pole through the southern Atlantic, Pacific and Indian Oceans back to South-America after 15 days (Section 7). This case is a perfect testbed for the ability of the model so simulate circumpolar movements, and evaluate its ability to represent the location of an aerosol plume after several days/weeks of travel.

### 3.1   Vertical mesh calculation

The vertical discretization of CHIMERE needs to obey twofold requirements. First, as it has been the case since the beginning of the development of the model, the vertical mesh needs to be very refined in the lowest atmospheric layers because these layers are critical for the modelling of boundary layer contamination, particularly in urban areas, but also in marine areas with sea-salt emissions, and in arid areas with mineral dust emissions. On the other hand, the CHIMERE model is now used not only for studies at urban/regional scale, but also for studies at continental and, from the present version, hemispheric scale. Therefore, a relatively fine vertical resolution is also needed in the free troposphere to be able to simulate the transport of trace gases and aerosols over large distances avoiding excessive numerical diffusion. Therefore, due to these two requirements, the CHIMERE-2017 vertical mesh is defined as described below.

Regarding the vertical discretization, the user has three degrees of freedom:

- the thickness of the first layer. The user can fix the top of the first model layer, by setting the top of the first model layer in sigma coordinates : $\sigma_1 = 0.997$ corresponds to a thickness of about 3 hPa for the first model layer, about 30 m.

- The number of layers, typically from 8 to 20 layers for the most common configurations of the model.

- The pressure of the top of the model, $p_{top}$, can be freely set by the user with typical values from 500 hPa for studies at urban/regional scales to 100 hPa for continental/hemispheric scale studies.

From these user-defined parameters, a preprocessing tool calculates a vertical grid as follows:

- From the surface to 800 hPa, the layer thickness (in hPa) increases exponentially

- From 800 hPa to the top of model, the layers are evenly distributed, with equal thickness for each layer.

This procedure outputs the pressure of the level tops, for a reference surface pressure $p_{ref}$ of 1000 hPa. However, the model levels need to adapt themselves to the variations of the surface pressure, essentially due to orography. This is ensured by scaling linearly the pressure levels between the surface pressure and the pressure at the top of model, $p_{top}$, producing two sequences of coefficients $a_i$ and $b_i$, such that the pressure at the top of level $i$ is given by $p_i = a_i p_{ref} + b_i p_{surf}$. These coefficients are given by the following expressions:

$$a_i = \frac{p_{top}\left(p_1 - p_i\right)}{p_{ref}\left(p_1 - p_{top}\right)} \tag{4}$$

$$b_i = \frac{p_1\left(p_i - p_{top}\right)}{p_{ref}\left(p_1 - p_{top}\right)} \tag{5}$$

The linear scaling of the pressure levels by these two sequences of coefficients ensures that the pressure levels never

cross each other, and that their relative thickness stays the same even above high topography, as shown in Fig. 4. Vertical transport on this mesh can be calculated using either a slope-limited Van Leer scheme (Van Leer, 1979) or a upwind scheme, depending on User's choice, also taking into account turbulent mixing and, optionnally, deep-convection fluxes, following the Tiedtke (1989) formulation.

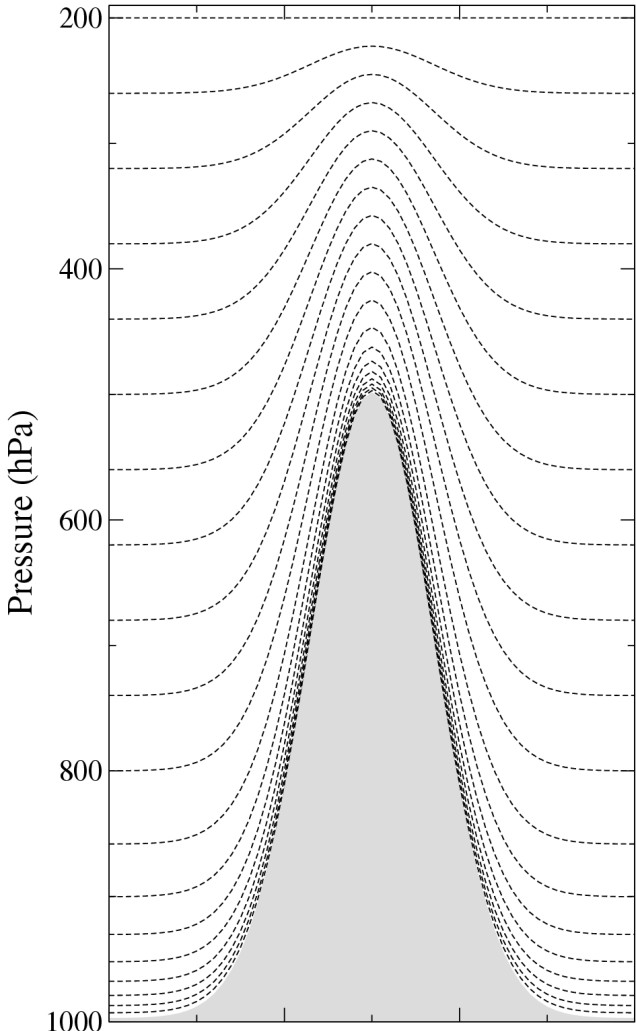

**Figure 4.** *Model pressure levels with 20 vertical levels, thickness of the first model layer is 3 hPa, top of model set at 200 hPa. Pressure levels are represented across an idealized mountain with a top at 500 hPa.*

## 4 Emissions

### 4.1 The anthropogenic emissions

#### 4.1.1 Overall description

CHIMERE needs to be forced at least by input meteorological fields, and by anthropogenic emissions. A preprocessor

for anthropogenic emissions, named *emisurf*, is provided to the users. This preprocessor was historically developed for the downscaling and reformatting of the raw emissions from the EMEP emission inventory at 50 km resolution, but can be adapted by users to any other raw dataset they need to use. The main steps for this are described in Menut et al. (2012):

– A first step projects the annual masses from the "raw" EMEP grid to the CHIMERE grid. The spatial emission distribution from the EMEP grid to the CHIMERE grid is performed using proxys like population density, as described by Figs. 5a-d. Proxies used by *emisurf* for this process include land use data (either GLCF, USGS or GlobCover), large point source database (such as the EPER database for Europe), etc.

– Second, monthly, weekly and hourly profiles are prescribed to convert annual totals to hourly fluxes used as input for CHIMERE. These factors are derived largely from data provided by the University of Stuttgart (IER) as part of the GENEMIS project (Friedrich and Reis, 2004), and are available as data files from the EMEP model website, www.emep.int.

– A last step consists in converting the species available in the raw data into the model species. Generally, a minimum of seven species are available: CO, $SO_x$, $NO_x$, $NH_3$, NMVOC, $PM_{2.5}$ and $PM_{coarse}$ (difference between $PM_{10}$ and $PM_{2.5}$). In CHIMERE, depending on the chemical scheme, about 30 species are emitted. $NO_x$ is split into NO, $NO_2$ and HONO. Usually, 5 to 10 % is assigned for $NO_2$ emissions for all sectors, except for traffic emissions where 20% should assigned to $NO_2$ for modern fleets (post 2010). For NMVOC, the VOC data used are derived from the detailed United Kingdom speciation given in Passant (2002). For $SO_x$, 99% is assigned to $SO_2$ and 1% for primary sulphate to account for very fast and local sulphate production. The lumping procedure accounts for the reactivity of VOC species following Middleton et al. (1990).

The vertical distributions were originally based upon plume-rise calculations performed for different types of emission sources which are thought typical for different emission categories, under a range of stability conditions (Vidic, 2002), but have since been simplified and adjusted to reflect the more recent findings of (Bieser et al., 2011). The main changes have been for the residential sector where now 100% of the emissions are placed in the lowest 20 m of the atmosphere, reflecting the large dominance of domestic combustion for this emission category. Also, emissions from large combustion facilities in SNAP ("Selected Nomenclature for Air Pollutants") sectors 1 and 4 corresponding to large industrial facilities burning fossil fuels are attributed to lower layers than in Vidic (2002), resulting in enhanced concentrations of primary species such as $NO_x$ and $SO_x$ in the

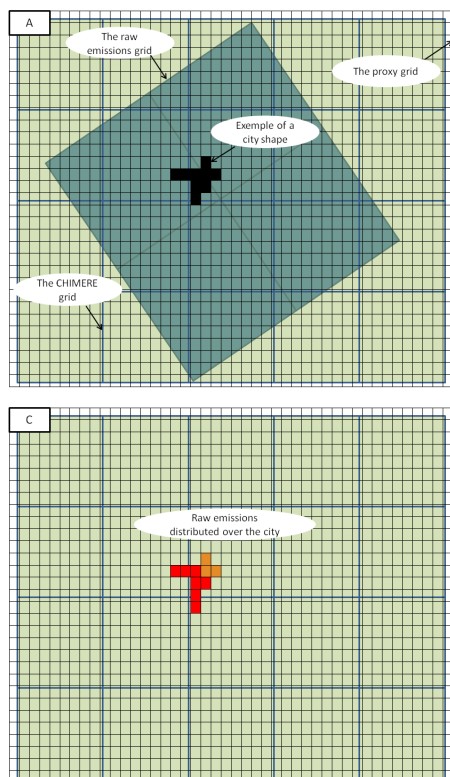
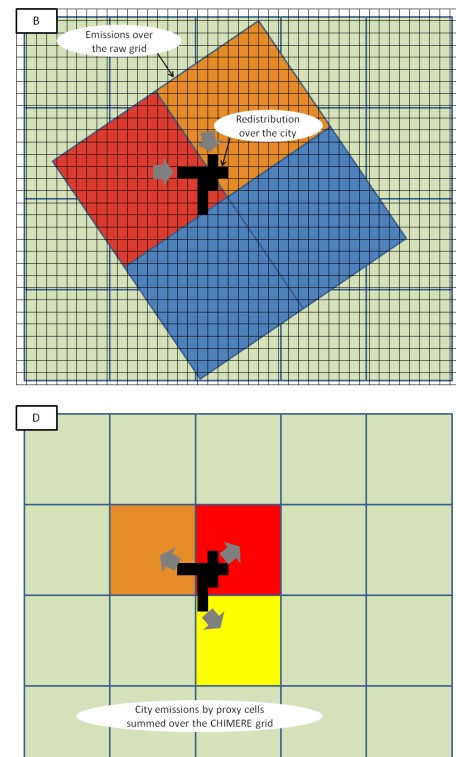

**Figure 5.** *Downscaling strategy for the anthropogenic emissions*

boundary layer, in better agreement with routine surface observations, as discussed in Mailler et al. (2013). The vertical distribution profiles that are used for each SNAP sector are constant profiles depending only on the SNAP sector, and are presented in Terrenoire et al. (2015)

### 4.1.2   Recent changes

The main recent changes have been focused on the use of proxies to better reallocate in space the raw emissions. This specialization can be performed from the raw gridded data or directly from the annual country totals (Terrenoire et al., 2015).

The European Pollutant Release and Transfer Register (E-PRTR) data are used to precisely place the emissions from the main industrial sources. E-PRTR is the Europe-wide register that provides easily accessible key environmental data from industrial facilities in European Union Member States and in Iceland, Liechtenstein, Norway, Serbia and Switzerland.

To treat road traffic emissions at the European scale, a spatial proxy to distribute the annual country emissions has been developed. This proxy provides a unitless value for a given cell at 1km resolution over Europe. It is built by crossing several databases (population, land cover data, roads, etc...), it consists of a linear regression of several parameters like

population density, length of road, and surface of urban areas in a given fine grid cell. The regression coefficients are calculated over France thanks to the use of the French high resolution bottom-up inventory and applied everywhere over Europe, Figure 6.

For the extrapolation at the European level, it uses the best source of information among the following proxies: CORINE land cover (from the European Environment Agency), road data of the ETISplus European project (European Transport policy Information System) for 2010 over Europe. ETISplus combines data, analytical modelling with maps (GIS), a single online interface for accessing the data. Default European GIS road data from EuroglobalMap, default worldwide GIS road data from Natural Earth data[1], and population database by Gallego (2010) over Europe and data from Center for International Earth Science Information Network (CIESIN) for the rest of the world. All of these data were not available on the whole domain. Therefore, three tiers of information were defined to cover all countries with different levels of confidence:

– Countries covered by all the data: Iceland, Norway, Turkey, Bosnia Herzegovinia, Serbia, Montenegro,

---

[1]http://www.naturalearthdata.com/

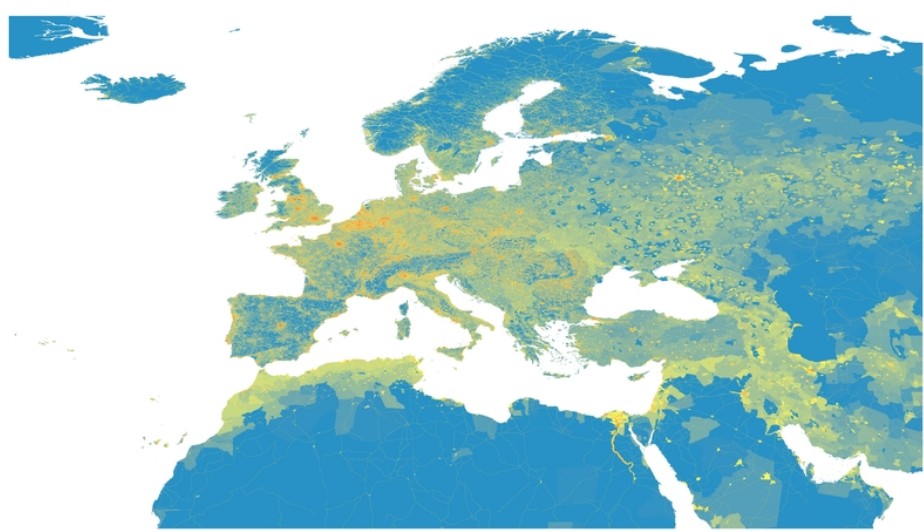

**Figure 6.** *Map of the unitless value calculated for the traffic emission proxy: low (blue) to high (red) values*

Kosovo, Macedonia, Albania and all the EU28 except Greece.

– Countries without CLC coverage but with ETIS or EuroglobalMap datas: Belarus, Ukraine, Moldavia and Greece

– Other countries are only covered by the world roadmap and population data.

For shipping emissions (SNAP 8), a proxy was developed using an inventory of shipping routes obtained from the US National Center for Ecological Analysis and Synthesis (NCEAS). A database of pressure on marine ecosystems has been developed for the year 2008 by Halpern et al. (2015) but the dataset remains non-exhaustive, the data being collected only on voluntary vessels.

## 4.2   Mineral dust emissions

Mineral dust modelling is an important process for understanding climate evolution but also for air quality regional modelling. For many regions over the world, it becomes necessary to manage air pollution knowing the relative part of anthropogenic and natural contributions. For this, even over small regions, it is important to have the same level of knowledge for mineral dust emissions as for anthopogenic or biogenic emissions. In this new model version, many improvements were done for mineral dust emissions. They are related to input databases, the emission schemes themselves and additional options to better take into account the impact of meteorological conditions on emissions.

### 4.2.1   Soil, landuse and roughness length

For the calculation of mineral dust emissions, several variables have to be known: landuse, soil characteristics, aeolian roughness length and erodibility. Originally, CHIMERE used a database limited to North Africa and the Arabian peninsula. For simulations over Africa or Europe, this spatially limited database was considered adequate, Sahara being the major source in this region. But for this new CHIMERE-2017 version, the goal is to enable calculations of mineral dust emissions anywhere in the world. It is then necessary to change from regional to global databases. A large part of this change was already done in Menut et al. (2013b) for landuse, soil and roughness length. The soil and landuse used are now those from NCAR USGS landuse dataset (Homer et al., 2004) and STATSGO-FAO soil dataset (Wolock, 1994). The roughness length is estimated using the global 6km horizontal resolution "Global Aeolian Roughness Lengths from ASCAT and PARASOL" dataset (Prigent et al., 2012).

In addition to these changes, the option to evaluate the soil erodibility based on satellite data was added. Therefore, three options are now available in CHIMERE 2017:

1. Calculate the erodibility from the landuse database: cropland, grassland, shrubland and barren or sparsely vegetated areas are then considered as partly erodible. This was the only option offered in earlier CHIMERE versions. In this case, constant percentages are applied for each landuse category.

2. Use the global erodibility dataset derived from MODIS (Grini et al., 2005), included and used in CHIMERE as described by Beegum et al. (2016).

3. Use a mix between these two strategies, using MODIS only over desert areas and the USGS landuses categories elsewhere.

### 4.2.2 The Kok's scheme for mineral dust emissions

In this model version, the Kok mineral dust emissions parameterization is proposed, in addition to the Marticorena and Bergametti (1995) and Alfaro and Gomes (2001) schemes.

The Kok scheme is fully described in the articles Kok et al. (2014b), Kok et al. (2014a) and Mahowald et al. (2014). The vertical dust flux is calculated as:

$$F_d = C_d\, f_{bare}\, f_{clay}\, \frac{\rho_a\left(u_*^2 - u_{*t}^2\right)}{u_{*st}} \left(\frac{u_*}{u_{*t}}\right)^{C_\alpha \frac{u_{*st} - u_{*st0}}{u_{*st0}}} \tag{6}$$

where $f_{bare}$ and $f_{clay}$ represent the relative fraction of bare soil and clay soil content, respectively. The flux is calculated only if $u_* > u_{*t}$. The threshold friction velocity, $u_{*t}$, is calculated using the Iversen and White (1982) or the Shao and Lu (2000) scheme (a user's choice). The corresponding $u_{*st}$ is this friction velocity but for a standard atmospheric density $\rho_{a0}$=1.225 kg $m^{-3}$:

$$u_{*st} = u_{*t}\sqrt{\frac{\rho_a}{\rho_{a0}}} \tag{7}$$

$u_{*st0}$ represents $u_{*st}$ for an optimally erodible soil and was chosen as $u_{*st0}$=0.16 m s$^{-1}$ in Kok et al. (2014b). The dimensionless coefficient $C_\alpha$ is chosen as $C_\alpha$=2.7.

The dust emission coefficient $C_d$ represents the soil erodibily as:

$$C_d = C_{d0}\exp\left(-C_e\frac{u_{*st} - u_{*st0}}{u_{*st0}}\right) \tag{8}$$

with the constant dimensionless coefficients $C_e$=2.0 and $C_{d0}$=4.4 $10^{-5}$.

The vertical dust flux is integrated over the whole size distribution. This flux is thus redistributed into the model dust size distribution as:

$$\frac{\mathrm{d}V_d}{\mathrm{d}\ln D_d} = \frac{D_d}{c_v}\left[1+\mathrm{erf}\left(\frac{ln(D_d/D_s)}{\sqrt{2}\ln\sigma_s}\right)\right]\exp\left[-\left(\frac{D_d}{\lambda}\right)^3\right] \tag{9}$$

with $V_d$ the volume of mineral dust aerosols for each mean mass median diameter $D_d$, $C_v$=12.62 $\mu$m, $\sigma_s$=3.0, $D_s$=3.4 $\mu$m and $\lambda$=12.0 $\mu$m.

### 4.2.3 Impact of vegetation on dust emissions

The vegetation evolves during the year and this variability will impact the mineral dust emissions. Contrarily to the previous model version, more focused on Saharan areas, this version is able to model mineral dust all around the world. For example in areas such as the Sahelian region or Europe, mineral dust are observed but are very dependent on the vegetation variability. To take into account this variability, the vegetation fraction is diagnosed from the USGS 30s resolution database and acts as a limiter to the erodibility factor.

### 4.2.4 Impact of rain on dust emissions

The possibility to inhibit or moderate dust erosion in case of rainfall was improved in this model version. In the previous model versions, the complete inhibition of mineral dust emissions during a rainfall event was already considered. In this version, a "rain memory function" was added in order to take into account the possible crustation of the soil (Ishizuka et al., 2008) and thus the fact that emissions are also reduced after a rainfall event. For this calculation, a simple factor $f_p$ is applied to moderate the dust emissions fluxes when a precipitation is diagnosed and during the next hours as:

$$f_p = E_{dust}\left(1 - \exp\left(\frac{-2\pi\,\Delta t_p}{\tau}\right)\right) \tag{10}$$

with $\Delta t_p$ the time since the last precipitation event and $\tau$ the period after which the surface mineral dust fluxes $E_{dust}$ is fully taken into account, considering that the inhibiting effect of precipitation is finished. For this study, $\Delta t_p$ is in hours and $\tau$=12. This function is displayed in Fig. 7.

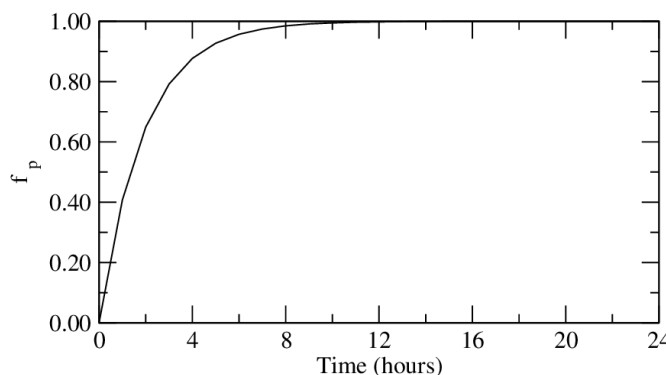

**Figure 7.** *Function defined to moderate the mineral dust emissions fluxes after a precipitation event.*

### 4.2.5 Impact of soil moisture on dust emissions

In the absence of precipitation, the soil moisture may also inhibit mineral dust erosion. This effect is taken into account using the Fecan et al. (1999) parameterization. This

scheme considers that soil moisture will increase the threshold friction velocity, $u_*^T$, used to determine if erosion occurs or not. To distinguish between soil conditions, the dry and wet threshold friction velocities are defined, and noted $u_*^{Td}$ and $u_*^{Tw}$, respectively. $u_*^{Tw}$ is estimated as a possible increase of $u_*^{Td}$ depending on the modelled gravimetric soil moisture $w$ (in kg.kg$^{-1}$):

$$u_*^{Tw} = f(w) \, u_*^{Td} \tag{11}$$

In the model, the dry treshold friction velocity, $u_*^{Td}$ is calculated following the scheme of Shao and Lu (2000). The $f(w)$ factor is estimated as:

$$
\begin{cases}
f(w) & = & 1 & \text{for } w < w' \\
f(w) & = & \left[1 + \text{A} \, (w - w')^{\text{b}'}\right]^{0.5} & \text{for } w > w'
\end{cases} \tag{12}
$$

where A and b$'$ are constants to estimate, and $w'$ corresponds to the minimum soil moisture from which the threshold velocity increases. The values of $A$, $b'$ and $w'$ are dependent on the soil texture. For $A$ and $b'$, the values are fixed to $A$=1.21 and $b'$=0.68. Using measurements data, Fecan et al. (1999) showed that the value of $w'$ is mainly dependent on the clay content of the soil and proposed the following fit:

$$w' = 0.0014(\%clay)^2 + 0.17(\%clay) \tag{13}$$

Note that in equation 12, the gravimetric soil moisture $w$ has to be expressed in %, $w'$ being in % in equation 13 (a conversion is done from kg/kg to %).

### 4.3 Traffic-related resuspension

The resuspension process is important for particulate matter and may induce a large increase of the emission flux in case of dry soils, for locations where traffic and industries produce particles that may be deposited on the ground and therefore become available for resuspension. In this model version, the resuspension flux is active only for cells containing an urbanized surface. This flux is applied as Primary Particulate Matter (PPM) emissions only and thus considered in the model as an anthropogenic process.

The formulation is derived from the bulk formulation originally proposed by Loosmore (2003). The resuspension rate $\lambda$, in s$^{-1}$, is expressed as:

$$\lambda = 0.01 \frac{u_*^{1.43}}{\tau^{1.03}} \tag{14}$$

where $\tau$ is the time after the start of resuspension. This time is taken into account considering that particles are first deposited then resuspended. The detail of the processes leading to resuspension are essentially unknown, and we assume

here that the available concentration of particulate matter depends only on the wetness of the surface. In this empirical view, the resuspension flux is assumed to be:

$$F = P \, f(w) \, u_*^{1.43} \tag{15}$$

where $f(w)$ is a function of the soil water content and $P$ is a constant tuned in order to approximately close the PM$_{10}$ mass budget over Europe estimated in Vautard et al. (2005). It was found to give a correct amount of additional $PM_{10}$. In this model version, $P$ is approximated as $P = 4.72 \, 10^{-2} \, \mu\text{gm}^{-2}\text{s}^{-1}$ if we consider European mean conditions with a soil water content of 25% and a friction velocity of $u_*$=0.5 m.s$^{-1}$.

The soil water function $f(w)$ is estimated as:

$$f(w) = \frac{w_s - w}{w_s - w_t} \tag{16}$$

where $w_t = 0.1$ is a soil moisture threshold below which resuspension is activated, and $w_s$ is the maximum of soil moisture ponderated by the ratio of water and soil densities, as:

$$w_s = w_{max} \frac{D_{water}}{D_{soil}} \tag{17}$$

with $w_{max} = 0.3$ is a constant value representing the maximum soil moisture value, $D_{water}$ is the water density (assumed to be unity) and $D_{soil}$ is the dry porous soil density. $D_{soil}$ is itself estimated as:

$$D_{soil} = (1 - satsm) \, D_{mine} \tag{18}$$

with $satsm$=0.4 the saturation volumetric moisture content and $D_{mine}$=2.5, the non-porous soil density.

This resuspension flux is calculated only for model cells having a non-zero urban landuse. This flux is thus ponderated in the whole cell by considering the relative surface of the urban area. Finally, the flux is projected onto the model size distribution considering that 2/3 of the flux is in the fine mode, 1/3 in the coarse mode. The fine and coarse modes are those defined for the anthropogenic emissions fluxes for particulate matter.

## 5 Processes and chemistry

### 5.1 Integration of the SAPRC chemical scheme

#### 5.1.1 The general gas-phase mechanism

Two gas-phase chemical schemes were implemented in the CHIMERE model. The most detailed chemical scheme, called MELCHIOR1, represents the oxidation of around

80 gaseous species according to 300 reactions. The other mechanism, called MELCHIOR2, is a reduced version of MELCHIOR1 developed using chemical operators (Derognat et al., 2003; Carter, 1990). MELCHIOR2 represents the oxidation of around 40 gaseous species according to 120 reactions. These chemical mechanisms are described in detail in Menut et al. (2013a). Comparisons between MELCHIOR2 and three detailed mechanisms (MCM, Jenkin et al. (2003) ; SAPRC99, Carter (2000) ; GECKO-A, Aumont et al. (2005)) show a good agreement between the chemical schemes, with differences in HCHO yields under low and high NO conditions lower than 20% between the simulated results (Dufour et al., 2009). SAPRC99 chemical mechanism had already been used in CHIMERE for particular studies (Lasry et al., 2007; Coll et al., 2009) but had never been distributed in a previous CHIMERE release.

Since the development of the MELCHIOR mechanisms in 2003, progress has been made in atmospheric chemistry, particularly concerning the VOC ozonolysis. One of the most up to date chemical schemes currently available in the literature is the SAPRC-07 (Carter, 2010a). This mechanism is widely used and evaluated against chamber data ($\approx$2400 experiments). The detailed SAPRC-07 chemical mechanism contains 207 species and 466 reactions. This detailed mechanism has been used to develop several reduced mechanisms designed for CTM applications (Carter, 2010b). The less reduced mechanism, SAPRC-07A, has been implemented in the 2016 CHIMERE model. This chemical scheme contains 72 species and 218 reactions. Two CHIMERE simulations using SAPRC-07A and MELCHIOR2 chemical schemes respectively were compared with AirBase measurements of $NO_x$ and ozone over Europe during summer 2005. The two chemical schemes were found to provide good correlation with ozone measurements (Pearson's correlation rate 0.71 for both mechanisms), with a slightly smaller bias for ozone concentrations obtained using SAPRC-07A (8.19 ppb versus 9.29 ppb, Menut et al. (2013a)).

### 5.1.2    The chlorine mechanism

Over the past decade, several studies have shown that halogens (chlorine, bromine, iodine) chemistry could influence ozone concentrations in the troposphere. A recent review by Simpson et al. (2015) presents the state of art on this topic.

The role of halogen chemistry was traditionally considered limited to the marine boundary layer, recent observations have shown significant $ClNO_2$ concentrations from few ppt in Mid-Continental Urban Environment (Mielke et al., 2011) to 2000 ppt in the Coastal Marine Boundary Layer (Riedel et al., 2011). This compound can act as a nitrogen reservoir with a long lifetime capable of long-range transport. In previous versions of CHIMERE, it was possible to have the chemical composition (Na, Cl, $H_2SO_4$) of sea salt emissions based on mean composition described in Seinfeld and Pandis (1997). The chlorine chemistry is not described in MELCHIOR chemical schemes but Carter (2010b) proposed in SAPRC-07A a chlorine mechanism with 9 inorganic species and 3 products formed by the reactions with VOCs. In SAPRC-07A, the chlorine chemistry is represented by 68 reactions, which have been implemented in CHIMERE-2017 only if the SAPRC-07A mechanism is chosen by the user.

## 5.2    Evolution of the aerosol scheme

### 5.2.1    Discretization of the aerosols size distribution

The CHIMERE model accounts for the size distribution of the aerosols using a size-bin approach: the aerosol particles for each of the model species are distributed in $N$ size bins, covering a diameter range from $D_{min}$ to $D_{max}$. Given these three user-defined parameters, a preprocessor computes a sequence $(d_i)_{i=1,N+1}$ of cutoff diameters that meets the following requirements:

– $2.5\,\mu$m and $10\,\mu$m are retained as cutoff diameters: two indices $i1$ and $i2$ such that $d_{i1} = 2.5\,\mu$m and $d_{i2} = 10\,\mu$m must exist

– The sequence of the cutoff diameters covers exactly the size interval requested by the user: $d_1 = D_{min}$ and $d_{N+1} = D_{max}$

The first requirement is set to allow a meaningful evaluation of $PM_{2.5}$ and $PM_{10}$ in the model, since these quantities are typically available from routine measurements.

The default (and recommended) values of the extreme diameters are $D_{min} = 0.01\,\mu$m and $D_{max} = 40\,\mu$m. Using these values, the produced size distributions for various values of the number of intervals $N$ are shown in Tab. 2 according to the requested number of bins, N. If $N \geq 12$, then the ratio of two successive cut-off diameters is always such as $d_{i+1}/d_i \leq 2$ : all particles within a single size bin have comparable diameters at least within a factor 2, which is a good way to ensure that all the size-depending processes affecting the aerosols (sedimentation, coalescence etc.) are treated in a realistic way. However, when calculation speed is a critical requirement, for example for operational prevision, the number of size bins could be lowered to $N = 6$, still ensuring that $d_{i+1}/d_i \leq 4$

### 5.2.2    Wet diameter and density of aerosols

In many processes, the diameter and the density of aerosols are used (deposition, absorption, coagulation, etc...). These processes have to take into account that the diameter and the density of aerosols change with humidity due to the amount of water absorbed into the particles. Therefore, the notion of wet diameter and wet density was introduced in CHIMERE-2017. Particles are distributed between bins according to their dry diameter. The wet diameter of the particles is calculated as a function of humidity and the composition of the particle.

| $I_v$ | Number of aerosol bins | | | | | | | | | | | | | |
|---|---|---|---|---|---|---|---|---|---|---|---|---|---|---|
| | N= 3 | N= 4 | N= 5 | N= 6 | N= 7 | N= 8 | N= 9 | N= 10 | N= 11 | N= 12 | N= 13 | N= 14 | N= 15 | N= 16 |
| 1 | 0.01 | 0.01 | 0.01 | 0.01 | 0.01 | 0.01 | 0.01 | 0.01 | 0.01 | 0.01 | 0.01 | 0.01 | 0.01 | 0.01 |
| 2 | 2.50 | 0.16 | 0.06 | 0.04 | 0.03 | 0.03 | 0.03 | 0.02 | 0.02 | 0.02 | 0.02 | 0.02 | 0.02 | 0.02 |
| 3 | 10.00 | 2.50 | 0.40 | 0.16 | 0.09 | 0.09 | 0.06 | 0.05 | 0.05 | 0.04 | 0.03 | 0.03 | 0.03 | 0.03 |
| 4 | 40.00 | 10.00 | 2.50 | 0.63 | 0.27 | 0.27 | 0.16 | 0.11 | 0.11 | 0.08 | 0.06 | 0.06 | 0.05 | 0.05 |
| 5 | / | 40.00 | 10.00 | 2.50 | 0.83 | 0.83 | 0.40 | 0.23 | 0.23 | 0.16 | 0.12 | 0.12 | 0.09 | 0.07 |
| 6 | / | / | 40.00 | 10.00 | 2.50 | 2.50 | 1.00 | 0.52 | 0.52 | 0.32 | 0.21 | 0.21 | 0.16 | 0.12 |
| 7 | / | / | / | 40.00 | 10.00 | 5.00 | 2.50 | 1.14 | 1.14 | 0.63 | 0.40 | 0.40 | 0.27 | 0.20 |
| 8 | / | / | / | / | 40.00 | 10.00 | 5.00 | 2.50 | 2.50 | 1.25 | 0.73 | 0.73 | 0.48 | 0.34 |
| 9 | / | / | / | / | / | 40.00 | 10.00 | 5.00 | 5.00 | 2.50 | 1.35 | 1.35 | 0.83 | 0.55 |
| 10 | / | / | / | / | / | / | 40.00 | 10.00 | 10.00 | 5.00 | 2.50 | 2.50 | 1.44 | 0.92 |
| 11 | / | / | / | / | / | / | / | 40.00 | 20.00 | 10.00 | 5.00 | 3.97 | 2.50 | 1.51 |
| 12 | / | / | / | / | / | / | / | / | 40.00 | 20.00 | 10.00 | 6.30 | 3.97 | 2.50 |
| 13 | / | / | / | / | / | / | / | / | / | 40.00 | 20.00 | 10.00 | 6.30 | 3.97 |
| 14 | / | / | / | / | / | / | / | / | / | / | 40.00 | 20.00 | 10.00 | 6.30 |
| 15 | / | / | / | / | / | / | / | / | / | / | / | 40.00 | 20.00 | 10.00 |
| 16 | / | / | / | / | / | / | / | / | / | / | / | / | 40.00 | 20.00 |
| 17 | / | / | / | / | / | / | / | / | / | / | / | / | / | 40.00 |

**Table 2.** *Values of the diameter intervals, $I_v$ (µm), obtained for $D_{min} = 0.01\,\mu$m, $D_{max} = 40\,\mu$m, and fourteen different values of bins (N=3 to N=16).*

To compute the wet density and wet diameter for each aerosol size bin, the amount of water in each bins is computed with the "reverse mode" of ISORROPIA (Nenes et al. (1998)) by using the composition of particles, assuming that only sulphate, nitrate, ammonium and sea salts have a high enough hygroscopicity to absorb a significant amount of water. The density of the aqueous phase of particles is computed according to composition following the method of Semmler et al. (2006). The density and mass of the inorganic aqueous phase (sulphate, nitrate, ammonium and sea salts and water) and the density and mass of other compounds (dust, organics, black carbon, etc...) are used to compute the total density of the particle and then its wet diameter, assuming internal mixing for each size bin.

### 5.2.3 Absorption

Absorption is described by the "bulk equilibrium" approach of Pandis et al. (1993). In this approach, all the bins for which condensation is very fast are merged into a "bulk particulate phase". Following Debry et al. (2007), a cutting diameter of $1.25\,\mu$m is used to separate bins which are inside the "bulk particle" (with a diameter lower than the cutting diameter) from other bins.

Thermodynamic models are used to compute the partitioning between the gas phase and the bulk particle phase and estimate the gas-phase concentrations at equilibrium. For semi-volatile inorganic species (sulphate, nitrate, ammonium), concentrations $G_{eq}$ at equilibrium are calculated using ISORROPIA. This model also determines the water content of particles. Equilibrium concentrations for the semivolatile organic species are related to particle concentrations through a temperature dependent partition coefficient $K^p$ (in m$^3\mu$g$^{-1}$) (Pankow (1994)).

Following Pandis et al. (1993), the mass of compounds condensing into particles, $\Delta A_p$, is redistributed over bins according to the kinetic of condensation into each bin. For evaporation, the mass of compounds evaporating from each bin is proportional to the amount of the compounds in the bin.

If the variation of particulate bulk concentration of compound i, $\Delta A_{p,i}$, is greater than 0 (condensation):

$$\Delta A_{p,i}^{bin} = \frac{k^{bin}}{\sum_j k_i^j} \Delta A_{p,i} \qquad (19)$$

with k$_i^{bin}$ the kinetic of condensation given by Seinfeld and Pandis (1997):

$$k_i^{bin} = N^{bin} \frac{2\pi D_p^{bin} D_i M_i}{RT} f(Kn, \alpha) \qquad (20)$$

with $N^{bin}$ the number of particles inside the bin, $D_p^{bin}$ the mean diameter of the bin, $D_i$ the diffusion coefficient for species i in air, $M_i$ its molecular weight and $f(Kn, \alpha)$ is the correction due to noncontinuum effects and imperfect surface accomodation. $f(Kn, \alpha)$ is computed with the transition regime formula of Fuchs and Sutugin (1971).

If the variation of particulate bulk concentration of compound i, $\Delta A_{p,i}$, is negative (evaporation):

$$\Delta A_{p,i}^{bin} = \frac{A_{p,i}^{bin}}{\sum_j A_{p,i}^j} \Delta A_{p,i} \qquad (21)$$

If a particle shrinks or grows due to condensation/evaporation, the mass of this particle has to be redis-

tributed over diameter bins. The mass redistribution algorithm of Gelbard and Seinfeld (1980); Seigneur (1982) is used.

### 5.2.4  Coagulation

The flux of coagulation $J^b_{coag,i}$ of a coumpound $i$ inside a bin $b$ is computed with the size binning method of Jacobson et al. (1994):

$$
J^b_{coag,i} = \sum_{j=1}^{b} \sum_{k=1}^{b} f^b_{j,k} K_{j,l} A^j_{p,i} N^k
$$
$$
- A^b_{p,i} \sum K_{b,j} N^k \tag{22}
$$

with $N^k$ the volumic number of particles in bin $k$, $K_{j,l}$ the coagulation kernel coefficient between bins $i$ and $j$ and $f^b_{j,k}$ the partition coefficient (the fraction of the particle created from the coagulation of bins j and k which is redistributed into bin b). The coagulation kernel and the partition coefficients are calculated as described in Debry et al. (2007).

### 5.2.5  Wet deposition

For the in-cloud scavenging of particles, the deposition of particles is assumed to be proportional to amount of water lost by precipitations. The deposition flux is written as:

$$
\left[ \frac{dQ^k_l}{dt} \right] = - \frac{\varepsilon_l P_r}{w_l h} Q^k_l \tag{23}
$$

with $P_r$ the precipitation rate released in the grid cell (kg m$^{-2}$ s$^{-1}$), w$_l$ the liquid water content (kg m$^{-3}$), $h$ the cell thickness (m) and $\varepsilon_l$ an empirical uptake coefficient (in the range 0 - 1) currently assumed to be 1. $l$ and $k$ are respectively the bin and composition subscripts.

For the below-cloud scavenging of particles, particles are scavenged by raining drops following Henzig et al. (2006). A polydisperse distribution of raining drops is applied:

$$
N(R) = 1.98 \, 10^{-5} \, A \, P^{-0.384} \, R^{2.93} \exp\left(-5.38 P^{-0.186} R\right) \tag{24}
$$

where

$$
A = 1.047 - 0.0436 \ln P + 0.00734 \left(\ln P\right)^2 \tag{25}
$$

with $P$ the precipitation rate in mm/h and $R$ the radius of the droplet. The below-cloud scavenging rate is written:

$$
\left[ \frac{dQ^k_l}{dt} \right] = -Q^k_l \int_R \pi R^2 u_g(R) E(R, r_l) N(R) \mathrm{d}R \tag{26}
$$

with $R$, the radius of the raindrop (in m), $r_l$ the radius of the particle (in m), $u_g$ the terminal drop velocity (in m/s), $E(R, r_l)$ the collision efficiency of a particle with a raindrop, $N(R)$ (in m$^{-4}$) the raindrop size distribution.

## 5.3  Online calculation of photolysis rates using the Fast-JX module

### 5.3.1  Modelling strategy

CHIMERE-2017 includes the module Fast-JX version 7.0b (Wild et al., 2000; Bian and Prather, 2002) for the online calculation of the photolysis rates. Fast-JX is a module which solves the equations of radiative transfer in an atmospheric column taking into account the Solar zenith angle, the vertical profile of ozone and water vapor concentrations, the ice- and water- clouds, the radiative effect of scattering and absorption by aerosols and the surface albedo.

Following the recommendations of the Fast-JX developers, the effective size of ice particles is estimated following Heymsfield (2003) as $\mathrm{Reff}_i = 164 \times IWC^{0.23}$, where $\mathrm{Reff}_i$ ($\mu$m) is the effective radius of ice particles, and $IWC$ is the ice content of the atmospheric particles (g.m$^{-3}$). Regarding water droplets, their radius is estimated also following the recommendations of fast-JX developers, as 9.60 $\mu$m for clouds at low altitudes (below 810 hPa), 12.68 $\mu$m for high clouds (above 610 hPa), and linearly interpolated between these two values for intermediate altitudes.

Taking these factors (and their real-time simulated variations) into account, Fast-JX computes the photolysis rates for all the relevant photochemical reactions that has been designed in order to be easily introduced in Chemistry-transport models, which has already been done in various CTMs such as PHOTOMCAT (Voulgarakis et al., 2009), Polair3D (Real and Sartelet, 2011), UKCA (Telford et al., 2013) and GEOS-Chem (Eastham et al., 2014).

CHIMERE-2013 did not take into account all of these processes (Menut et al., 2013a), relying instead on a very simplified calculation of the photolysis rates, as shown in Table 3. The photolysis rates were evaluated from tabulated values using TUV (Madronich, 1987), depending only on the solar zenith angle and the altitude. These tabulated values were calculated assuming a vertical profile for ozone that was typical of the northern hemisphere midlatitudes, neglecting the effect of the aerosols, and assuming a constant and uniform surface albedo. The effect of clouds was parameterized as an exponential reduction of the photolysis rates as a function of the cloud optical depth. While this set of approximations was acceptable when the CHIMERE model was used as boundary-layer regional CTM for locations in Europe, this had strong limitations for its use for longer-term simulations including long-range transport in the free troposphere over geographical domains including polar and/or tropical zones. Photolysis rates for the photodissociation of ozone and nitrogen dioxide as computed by the Fast-JX model inside CHIMERE have been compared favorably to in situ measurements at the island of Lampedusa (Italy), even in presence of aerosols (Mailler et al., 2016)

|                              | CHIMERE-2013      | CHIMERE-2017                            |
|------------------------------|-------------------|-----------------------------------------|
| SZA                          | ✓                 | ✓                                       |
| Altitude                     | ✓                 | ✓                                       |
| Clouds                       | parameterized     | ✓                                       |
| Tropospheric ozone column    | Constant profile  | ✓                                       |
| Stratospheric ozone column   | Constant profile  | Month- and latitude-dependant climatology |
| Water-vapor concentration    | Constant profile  | ✓                                       |
| Aerosol effect               | ✗                 | ✓                                       |
| Variable albedo              | ✗                 | ✓                                       |

**Table 3.** *Taking into account the various factors affecting the photolysis rates in CHIMERE-2013 and CHIMERE-2017*

### 5.3.2   surface albedo

The surface albedo in the near-UV spectral region, which is determinant for the calculation of photolysis rates (Dickerson et al., 1982), is highly variable according to the landuse and to the presence or absence of snow. It is worth noting that the albedo of all the continental and oceanic surfaces is smaller than 0.1, while the albedo of snow ranges from 0.3 to over 0.8 according to the type of landuse. Therefore, the absence/presence of snow will modulate very substantially the values of the modelled photolysis rates, and therefore the concentration of trace gases such as ozone. Even though strong ozone peaks generally occur in summertime in a context of strong anthropogenic $NO_x$ production and in the absence of snow, it has been shown recently that strong ozone peaks can occur in wintertime over the continental United States in zones of oil and gas extraction due to the combination of the strong anthropogenic concentrations of VOCs in a very shallow boundary layer with relatively strong photolysis rates due to the high surface albedo (Edwards et al., 2014; Schnell et al., 2009). It is therefore important that CTMs take into account the impact of snow on surface albedo, in order to be able to reproduce correctly such cases.

The surface albedo in the UV band in CHIMERE-2017 is evaluated according to Laepple et al. (2005) in the absence of snow (tested as snow depth less than 1 cm), and from Tanskannen and Manninen (2007) in the presence of snow, tested as snow depth greater than 10 cm. Values are displayed in Table 4.

The snow depth is read from the WRF or ECMWF meteorological inputs, if available. If any other model is used, the snow cover will be assumed inexistent. If the snow-cover is thinner than 1 cm in the model, the albedo is assumed to be that of dry land. If the snow-cover is thicker than 10 cm, the albedo is assumed to be that of snow-covered land. In-between, a linear interpolation is performed. Even though the case of sea-ice is not explicitly treated in Tanskannen and Manninen (2007), the assumption is made in CHIMERE-2017 that the albedo of sea-ice is the same as that of a thick layer of snow covering barren land.

| #  | Landuse                   | albedo for snow | |
|----|---------------------------|-----------------|---------|
|    |                           | < 1cm           | > 10 cm |
| 1  | Agricultural land / crops | 0.035           | 0.376   |
| 2  | GrasslandLanduse type     | 0.04            | 0.720   |
| 3  | Barren land/bare ground   | 0.10            | 0.836   |
| 4  | Inland Water              | 0.07            | -       |
| 5  | Urban                     | 0.035           | 0.3     |
| 6  | Shrubs                    | 0.05            | 0.558   |
| 7  | Needleaf forest           | 0.025           | 0.278   |
| 8  | Broadleaf forest          | 0.025           | 0.558   |
| 9  | Ocean                     | 0.07            | 0.836   |

**Table 4.** *Tabulated values from Laepple et al. (2005) and Tanskannen and Manninen (2007) used for the calculation of the albedo in the UV band. In the presence of sea-ice over ocean, the albedo of the ice surface is assumed equal to the Tanskannen and Manninen (2007) value for > 10 cm of snow on barren land.*

### 5.3.3   Implementation

The physical calculations performed by Fast-JX are split in two steps.

First, the Legendre coefficients for the scattering phase function for all aerosol species and diameter bin are calculated using Michael Mischenko's spher.f code (Mischenko et al., 2002), assuming sphericity of the aerosol particles. This calculation is performed for each of the $n_{spec} \times n_{bins}$ species, and for the five wavelengths that are used for the Mie scattering processes in Fast-JX. This step is performed once and for all before the first simulation step, and lasts from a couple of seconds to a couple of minutes according to the number of aerosol species and diameter bins. The refractive indices reproduced in Table 5 are the ones provided along with the model, essentially based on the values compiled in the framework of the ADIENT project[2], as described by the corresponding technical report by E. J. Highwood[3]. However, the specification of these parameters is in a param-

---

[2]http://www.reading.ac.uk/adient/refractiveindices.html, visited Jan. 17, 2017

[3]www.reading.ac.uk/adient/REFINDS/Techreportjul09.doc, visited Jan. 17, 2017

eter file, and can be changed by the user to other values. In the same way, the user can easily introduce more species in the optical treatment for specific studies, e.g. volcanic ashes.

After the preprocessing phase, at each time step and in each model column, the Fast-JX module resolves the radiative transfer in the model atmospheric column, computing the actinic fluxes at each model level and integrating them over $N$ wavelength bins in order to produce accurate photolysis rates. In the configuration adopted for CHIMERE-2017, $N$ is set to 12, which is the value recommended by Fast-JX developers for tropospheric studies. These 12 wavelength bins include the 7 standard Fast-J wavelength bins from 291 nm to 850 nm, as described in Wild et al. (2000). The 7 standard Fast-J wavelength bins are essentially concentrated from 291 nm to 412.5 nm which is the spectral band relevant for tropospheric photochemistry. Following the recommendations of Fast-JX model developers, these 7 standard wavelength bins are complemented by 5 additional wavelength bins, from 202.5 nm to 291 nm, which are only relevant in the upper tropical troposphere. In a typical simulation framework, it has been found that the increase in computational time relative to the simulation with tabulated photolysis rates is below 10% (Mailler et al., 2016).

### 5.4 Online calculation of lidar profiles

During the model integration, some additional diagnostic variables are estimated: (i) the Clouds Optical depth (COD) and the Aerosol Optical Depth (AOD) using the FastJX module, and (ii) the lidar profiles.

The lidar profiles are calculated using the aerosol contributions only, as detailed in Stromatas et al. (2012). They are proposed as output after a simulation and are designed to be directly comparable to ground-based or spatial lidars. Three different profiles are calculated both in Nadir and Zenith lidar configurations: (i) the Attenuated Scattering Ratio, $R'(z)$, (ii) $\beta'(z,\lambda)$ and $\beta'_{\mathrm{m}}(z,\lambda)$, respectively the total and molecular attenuated backscatter signal.

By definition, $R'(z)$ is equal to 1 in absence of aerosols/clouds and when the signal is not attenuated. In the presence of aerosols, $R'(z)$ would be greater than one. Following Winker et al. (2009), this ratio is expressed as:

$$R'(z) = \frac{\beta'(z)}{\beta'_{\mathrm{m}}(z)} \tag{27}$$

The total attenuated backscatter signal $\beta'(z,\lambda)$ is calculated as:

$$
\begin{aligned}
\beta'(z,\lambda) \quad = \quad & \left[ \frac{\sigma_{\mathrm{m}}^{\mathrm{sca}}(z,\lambda)}{S_{\mathrm{m}}(z,\lambda)} + \frac{\sigma_{\mathrm{p}}^{\mathrm{sca}}(z,\lambda)}{S_{\mathrm{p}}(z,\lambda)} \right] \\
& \exp \left( -2 \left[ \int_{z}^{\mathrm{TOA}} \sigma_{\mathrm{m}}^{\mathrm{ext}}(z',\lambda)\mathrm{d}z' \right. \right. \\
& \left. \left. + \eta' \int_{z}^{\mathrm{TOA}} \sigma_{\mathrm{p}}^{\mathrm{ext}}(z',\lambda)\mathrm{d}z' \right] \right)
\end{aligned}
\tag{28}
$$

and the molecular attenuated backscatter signal $\beta'_{\mathrm{m}}(z,\lambda)$ as:

$$\beta'_{\mathrm{m}}(z,\lambda) = \frac{\sigma_{\mathrm{m}}^{\mathrm{sca}}(z,\lambda)}{S_{\mathrm{m}}(z,\lambda)} \cdot \exp \left( -2 \int_{z}^{\mathrm{TOA}} \sigma_{\mathrm{m}}^{\mathrm{ext}}(z',\lambda)\mathrm{d}z' \right) \tag{29}$$

$\sigma_{\mathrm{p}}^{\mathrm{sca/ext}}(z,\lambda)$ and $\sigma_{\mathrm{m}}^{\mathrm{sca/ext}}(z,\lambda)$ are the extinction/scattering coefficients for particles and molecules (in km$^{-1}$). $S_{\mathrm{m}}$ and $S_{\mathrm{p}}$ are the molecular and particular extinction-to-backscatter ratios (in $sr$). $\eta'(z)$ represents the particles multiple scattering and $z$ represents the distance between the emitter and the studied point. Note that for the case of a space lidar the integration begins from the top of the atmosphere (TOA) while for a ground lidar the integration begins from 0 (ground level) to $z$. Further details about these calculations are provided in Stromatas et al. (2012).

## 6 Model scores for two test cases over Europe

The performance of CTMs is often evaluated by comparing simulation results to data of measurements, either from routine networks (Solazzo et al., 2012a, b) or from dedicated field campaigns (e.g. Menut et al. (2015); Petetin et al. (2015)). Simon et al. (2012) presented an overview of performance evaluation studies for a large set of models and studied cases.

A statistical evaluation with measurement data is performed for two 3-months long simulations with CHIMERE-2017: summer (June to August 2008) and winter (January to March 2009). Each of the simulation periods analyzed was preceeded by a 15-day spinup period. The simulation domain covers western and central Europe at 0.5° resolution, with 8 vertical sigma levels between 997 and $500\,\mathrm{hPa}$. The meteorological model used was WRF 3.6.1 with the same physical options as in (Menut et al., 2015), xpat 45 km resolution and boundary conditions from GFS analyses. The emission data were those from EMEP at 0.5°, and the boundary conditions for the concentrations from the LMDz-INCA model for gases and chemically active aerosols and from the GOCART model for dust. The simulation was performed with

| Species | Real part of the refractive index | | | | | Imaginary part of the refractive index | | | | |
|---|---|---|---|---|---|---|---|---|---|---|
| $\lambda$ | 200 nm | 300 nm | 400 nm | 600 nm | 1000 nm | 200 nm | 300 nm | 400 nm | 600 nm | 1000 nm |
| PPM | 1.53 | 1.52 | 1.52 | 1.51 | 1.50 | $8.0\cdot10^{-3}$ | $8.0\cdot10^{-3}$ | $8.0\cdot10^{-3}$ | $8.0\cdot10^{-3}$ | $8.0\cdot10^{-3}$ |
| OCAR | 1.60 | 1.60 | 1.63 | 1.63 | 1.63 | $1.2\cdot10^{-1}$ | $1.2\cdot10^{-1}$ | $7.7\cdot10^{-2}$ | $1.2\cdot10^{-2}$ | $7.0\cdot10^{-2}$ |
| BCAR | 1.85 | 1.85 | 1.85 | 1.85 | 1.85 | $7.1\cdot10^{-1}$ | $7.1\cdot10^{-1}$ | $7.1\cdot10^{-1}$ | $7.1\cdot10^{-1}$ | $7.1\cdot10^{-1}$ |
| SALT | 1.38 | 1.38 | 1.37 | 1.36 | 1.35 | $8.7\cdot10^{-7}$ | $3.5\cdot10^{-7}$ | $6.6\cdot10^{-9}$ | $1.2\cdot10^{-8}$ | $2.6\cdot10^{-5}$ |
| SOA | 1.56 | 1.56 | 1.56 | 1.56 | 1.56 | $3.0\cdot10^{-3}$ | $3.0\cdot10^{-3}$ | $3.0\cdot10^{-3}$ | $3.0\cdot10^{-3}$ | $3.0\cdot10^{-3}$ |
| DUST | 1.53 | 1.53 | 1.53 | 1.53 | 1.53 | $5.5\cdot10^{-3}$ | $5.5\cdot10^{-3}$ | $2.4\cdot10^{-3}$ | $8.9\cdot10^{-4}$ | $7.6\cdot10^{-4}$ |
| H2SO4 | 1.50 | 1.47 | 1.44 | 1.43 | 1.42 | $1.0\cdot10^{-8}$ | $1.0\cdot10^{-8}$ | $1.0\cdot10^{-8}$ | $1.3\cdot10^{-8}$ | $1.2\cdot10^{-6}$ |
| HNO3 | 1.53 | 1.53 | 1.53 | 1.53 | 1.53 | $6.0\cdot10^{-3}$ | $6.0\cdot10^{-3}$ | $6.0\cdot10^{-3}$ | $6.0\cdot10^{-3}$ | $6.0\cdot10^{-3}$ |
| NH3 | 1.53 | 1.52 | 1.52 | 1.52 | 1.52 | $5.0\cdot10^{-4}$ | $5.0\cdot10^{-4}$ | $5.0\cdot10^{-4}$ | $5.0\cdot10^{-4}$ | $5.0\cdot10^{-4}$ |
| WATER | 1.35 | 1.34 | 1.34 | 1.33 | 1.33 | $2.0\cdot10^{-9}$ | $2.0\cdot10^{-9}$ | $1.8\cdot10^{-8}$ | $3.4\cdot10^{-8}$ | $3.9\cdot10^{-7}$ |

**Table 5.** *Refractive indices for the main aerosol species in CHIMERE at 200, 300, 400, 600 and 1000 nm*

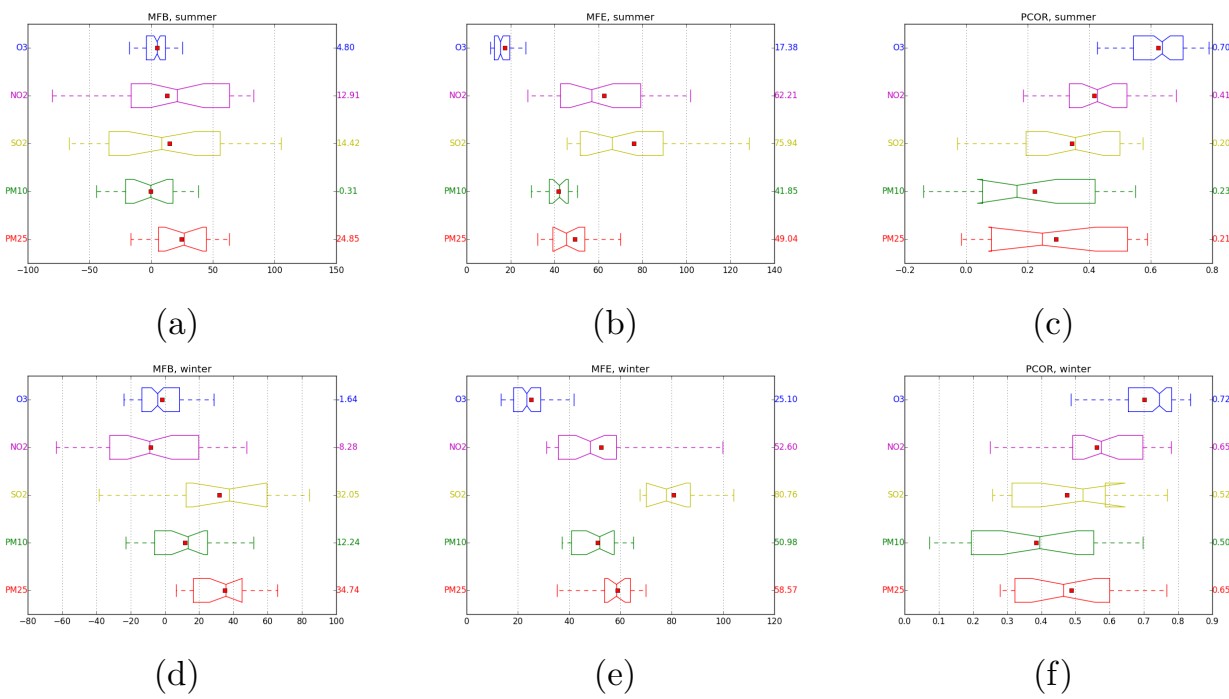

**Figure 8.** *Performance statistics for the main model species and for daily averaged values. The numbers on the right axes give the overall scores (Pearson's correlation, MFE, and MFB), while the box plots show the variability among the EMEP stations. The boxes extend from the lower to upper quartile values of the data. The center lines show the medians, and the red squares show the means over stations. The whiskers indicate the 5 and 95 percentile values, and the values on the right axis of each panel are the overall value of the considered indicator, i.e. merging all the stations into a single statistical dataset as described in Jacobson (2005)*

the MELCHIOR2 chemical mechanism for gaseous species, 10 bins for aerosol size distribution and the SOA scheme of Bessagnet et al. (2008), 5-min chemistry time step, and the Van Leer numerical scheme for both horizontal and vertical transport. The (Wesely, 1989) aerosol dry deposition and (Loosmore, 2003) resuspension schemes were used. The on-line coupling with ISORROPIA model was used.

The statistical scores are computed between modelled and observed daily averaged values, using surface concentration measurements from the EMEP modeeting sites, after filtering out the stations with complex topography (CH01 CH04 CH05 DE03 DE08 AT05 AT48 IT01 IT04 ES78 DE44) that cannot be simulated appropriately at 0.5° resolution. Stations from the EMEP monitoring sites have been chosen for this study because their location has been selected in order to minimize local influences and be representative of large areas (Tørseth et al., 2012). For each simulation period only

| Species | Winter | Summer |
|---------|--------|--------|
| O3      | 96     | 93     |
| NO2     | 40     | 34     |
| SO2     | 12     | 27     |
| PM10    | 26     | 20     |
| PM25    | 22     | 16     |

**Table 6.** *Number of EMEP stations per species and per season used for performance statistics. Stations CH01 CH04 CH05 DE03 DE08 AT05 AT48 IT01 IT04 ES78 DE44 were excluded from the analysis due to their topography difficult to simulate with a 0.5° resolution.*

stations containing at least 70% of time series data were retained.

Figure 8 shows the performance statistics for the main model species. The number of EMEP stations used for each species for winter and summer is shown in Table 6. The standard metrics used for air quality modeling (Simon et al., 2012) were employed, namely the Pearson's correlation $PCOR$, the mean fractional error $MFE$, and the mean fractional bias $MFB$.

Ozone shows the best scores among all the species, both for summer and winter, with $PCOR = 0.70$, $MFE = 17\%$, $MFB = 5\%$ in summer and $PCOR = 0.72$, $MFE = 25\%$, $MFB = 2\%$ in winter. It also shows the smallest variability of scores among the stations (93 available stations in summer and 96 in winter). As noted by Simon et al. (2012), the ozone overestimation often reported for CTMs is related to the averaging over the hours with high and low concentrations, so the scores are dominated by performance at low concentrations, which occur much more often than high concentrations. Indeed, the MFB computed from daily maximum ozone concentrations (not shown) is quite lower: 1% for summer and 7% for winter.

The NO$_2$ shows quite larger MFE: 62% in summer and 53% in winter, with a large variability of both MFE and MFB between stations. The bias is negative in winter, slightly positive in summer but with a high negative values (NO$_2$ underestimation) at some stations. For this particular species, with strong emissions horizontal gradients, the model resolution of 0.5 ° is not enough even when surface concentrations are measured at the background rural sites. Also, as discussed by Terrenoire et al. (2015), the negative bias could be partly related to the general underestimation of the emissions in the inventory used, especially during the traffic daily peaks. This is in agreement with the relatively high correlation: 0.65 in winter and 0.41 in summer. However, this would not explain why there is a small positive bias in summer for most stations.

The SO$_2$ shows the largest MFE for both summer (76%) and winter (81%) and the lowest correlation in summer (0.20). It shows positive bias: $MFB =$32% in winter and 14% in summer. The difficulty in SO$_2$ simulation could be related to the uncertainties in the emission vertical profiles,

which is a particularly sensitive factor in SO$_2$ modelling, because industrial stack emissions represent a substantial part of SO$_2$ emissions (Pirovano et al., 2012; Mailler et al., 2013). While some CTMs have included a plume-in-grid model for subgrid treatment of point emissions depending on the actual meteorological conditions and flux characteristics, this is not the case of the CHIMERE model, which can also limit the performance of the model regarding SO$_2$ concentrations. The conversion of SO$_2$ to sulphate can also be a source of error in SO$_2$ concentrations, as mentioned by Ciarelli et al. (2016) and Bessagnet et al. (2016), who observed very different behaviour of models far from emission sources, probably due to the chemical mechanisms. The lower correlation coefficient in summertime was found in all the CTMs examined in Bessagnet et al. (2016).

The performance for PM is affected by compensating effects of several chemical components, such as dust, primary organics and secondary species like sulphates, nitrates, and SOA.

The PM$_{10}$ concentrations are generally overestimated in winter ($MFB = 12\%$), with correlation values lower in winter (0.50) and summer (0.23) than for the whole year, as reported by Terrenoire et al. (2015). In summer the PM$_{10}$ bias is quite low $MFB \approx 0\%$, and the $MFE$ (42%) shows small variability between the stations.

The PM$_{25}$ concentrations show a larger overestimation than PM$_{10}$ in winter ($MFB = 35\%$ vs 12% for PM$_{10}$) and have also a positive bias in summer ($MFB = 25\%$). The winter correlation is higher though (0.65 vs 0.50), and its variability between the stations is smaller. The PM$_{25}$ overestimation can be associated to the overestimation of ammonium ($MFB =$77% in summer and 65% in winter) and sulphate ($MFB =$32% in summer and 33% in winter, not shown).

Boylan and Russell (2006) define performance goals and criteria to be attained by air quality models. Their performance goal is attained for particulate matter when the MFE is less or equal to 50%, and |MFB| is less than 30%. The performance criteria are attained when the MFE is less or equal to 75%, and |MFB| is less than 60%. The performance goal is thus a more demanding condition than the performance criteria.

The PM$_{10}$ simulation satisfies the performance goal for both summer and winter. As for PM$_{25}$, it satisfies the performance goal in summer and the performance criteria in winter.

## 7  Application to the Puyehue-Cordon Caulle eruption (June 2011)

A simulation with the present version of CHIMERE has been performed for the southern hemisphere, from May 15 to June 30, 2011, a period covering the eruption of Puyehue-Cordon Caulle (Chile). This eruption has emitted an important plume containing volcanic ashes and sulphur dioxide into the tropo-

sphere and the lower stratosphere. This plume has had severe consequences on air traffic over Argentina as well as other countries in the southern hemisphere. While the eruption began on June 4, the plume went around the entire southern hemisphere and was back in the vicinity of the emission source by June 14 (Global Volcanism Program, 2013; Klüser et al., 2013). This volcanic eruption case provides a perfect testbed to evaluate the new abilities of the CHIMERE model to simulate as accurately as possible transport at hemispheric scale, including cases where the transported plume undergoes a complete circumpolar trajectory around the South Pole.

### 7.1   Model configuration

The meteorological simulation has been performed using the WRF meteorological model, version 3.5.1, on a simulation domain covering most of the southern hemisphere at a resolution of about 55 x 55 km at 45°S. with 20 vertical levels from the surface to 100 hPa. For the gasous chemistry, the MELCHIOR-2 chemical mechanism has been used. The horizontal domain is composed of 250x250 cells and is centered at the south pole and covering the entire extratropical southern hemisphere. The horizontal resolution varies with latitude: 65km x 65km (at the pole), 55 x 55 km (at 45°S), 36 x 36 km at at 25°S.

The anthropogenic and biogenic emissions are taken into account and produced from the HTAP dataset and MEGAN model, respectively. Mineral dust emissions have not been included in this simulation, since the focus of this testbed study was in the circumpolar transport of ash emissions from the Puyehue volcano. The novely of this simulation is the addition of the volcanic emissions of $SO_2$ and volcanic ashes.

### 7.2   Volcanic emissions

The total mass flux emitted in the form of particles has been represented according to Mastin et al. (2009), using the following equation,

$$\dot{V} = \left(\frac{H}{2.00}\right)^{\frac{1}{2.41}}$$
$$\dot{M} = \rho\dot{V} \tag{30}$$

where $H$ is the column height expressed in km, $\dot{V}$ is the volume flux expressed in $m^3 s^{-1}$, $\dot{M}$ is the mass flux in $kg\,s^{-1}$ and $\rho = 2500\,kg\,m^{-3}$ is the ash density. The altitude of the ash column has been taken from Collini et al. (2013), and is reproduced here in Table 7. Only the fine fraction of the emissions, with particle diameter smaller than $63\,\mu m$ has been included. The conversion from the total emitted mass flux has been performed using a conversion factor $m_{63}$ taken from Mastin et al. (2009) for S2 type volcanoes, i.e. $m_{63} = 0.4$. It is worth noting at this point that the uncertainty on the value of this parameter, $m_{63}$, is very strong, with values ranging

| day | H | $\dot{V}$ | $\dot{M}$ | M | $M_{63}$ |
|---|---|---|---|---|---|
| 04/06 | 10 | 794.9 | $1.99 \times 10^{06}$ | $2.86 \times 10^{10}$ | $1.14 \times 10^{10}$ |
| 05/06 | 10 | 794.9 | $1.99 \times 10^{06}$ | $1.72 \times 10^{11}$ | $6.87 \times 10^{10}$ |
| 06/06 | 10 | 794.9 | $1.99 \times 10^{06}$ | $1.72 \times 10^{11}$ | $6.87 \times 10^{10}$ |
| 07/06 | 6.5 | 133.0 | $3.33 \times 10^{05}$ | $2.87 \times 10^{10}$ | $1.15 \times 10^{10}$ |
| 08/06 | 7 | 180.9 | $4.52 \times 10^{05}$ | $3.91 \times 10^{10}$ | $1.56 \times 10^{10}$ |
| 09/06 | 8.5 | 405.0 | $1.01 \times 10^{06}$ | $8.75 \times 10^{10}$ | $3.50 \times 10^{10}$ |
| 10/06 | 8 | 314.9 | $7.87 \times 10^{05}$ | $6.80 \times 10^{10}$ | $2.72 \times 10^{10}$ |
| 11/06 | 6.5 | 133.0 | $3.33 \times 10^{05}$ | $2.87 \times 10^{10}$ | $1.15 \times 10^{10}$ |
| 12/06 | 7 | 180.9 | $4.52 \times 10^{05}$ | $3.91 \times 10^{10}$ | $1.56 \times 10^{10}$ |
| 13/06 | 8 | 314.9 | $7.87 \times 10^{05}$ | $6.80 \times 10^{10}$ | $2.72 \times 10^{10}$ |
| 14/06 | 7 | 240.9 | $6.02 \times 10^{05}$ | $5.20 \times 10^{10}$ | $2.08 \times 10^{10}$ |
| 15/06 | 8 | 314.9 | $7.87 \times 10^{05}$ | $6.80 \times 10^{10}$ | $2.72 \times 10^{10}$ |
| 16/06 | 7 | 180.9 | $4.52 \times 10^{05}$ | $3.91 \times 10^{10}$ | $1.56 \times 10^{10}$ |
| 17/06 | 5.5 | 66.5 | $1.66 \times 10^{05}$ | $1.44 \times 10^{10}$ | $5.75 \times 10^{09}$ |
| 18/06 | 5 | 44.8 | $1.12 \times 10^{05}$ | $9.68 \times 10^{09}$ | $3.87 \times 10^{09}$ |
| 19/06 | 4 | 17.7 | $4.44 \times 10^{04}$ | $3.83 \times 10^{09}$ | $1.53 \times 10^{09}$ |
| 20/06 | 4 | 17.7 | $4.44 \times 10^{04}$ | $3.83 \times 10^{09}$ | $1.53 \times 10^{09}$ |

**Table 7.** *Main characteristics of the volcanic emissions used for the hemispheric simulation. H : column height (km) ; $\dot{V}$ : volume flux ($m^3 s^{-1}$); $\dot{M}$ : Mass flux ($kg\,s^{-1}$); M : emitted mass (kg); $M_{63}$ : emitted mass for the fraction with diameter $< 63\mu m$*

from 0.02 to 0.6 depending on the characteristics of the considered eruption, and that therefore the uncertainties on the resulting mass of fine ash is very strong. The particles emitted with a diameter greater than $63\mu m$ have not been considered because they are not supposed to be relevant for long-range transport due to their rapid sedimentation.

The emitted ashes have been distributed evenly from the altitude of the crater (2200 m.a.s.l) to the altitude of the top of the column, obtained by summing the column height to the altitude of the crater.

The refractive indices of the volcanic ashes from Derimian et al. (2012) have been used. However, as these authors provide the refractive indices of volcanic ash only in the visible, the values at 200 and 300 nm have been taken as equal to the value given at 440 nm.

The granulometry of the ashes are taken as 80% in a coarse mode, with a lognormal distribution centered at $30\,\mu m$ and 20% in a finer mode with a lognormal distribution centered at $4\,\mu m$, consistent with the results of Durant et al. (2009).

The $SO_2$ mass flux has been taken from Theys et al. (2013), who prescribe mass flux estimates based on IASI measurements for the first 48 hours of the eruption. Since these authors do not provide an estimation for the subsequent part of the eruption, we assumed that the $SO_2$ fluxes are null after the 48 first hours of the eruption. This hypothesis is of course questionable, but nevertheless the study of Theys et al. (2013) shows in a convincing way that most of the $SO_2$ emission occurs during the first 48 hours of the eruption.

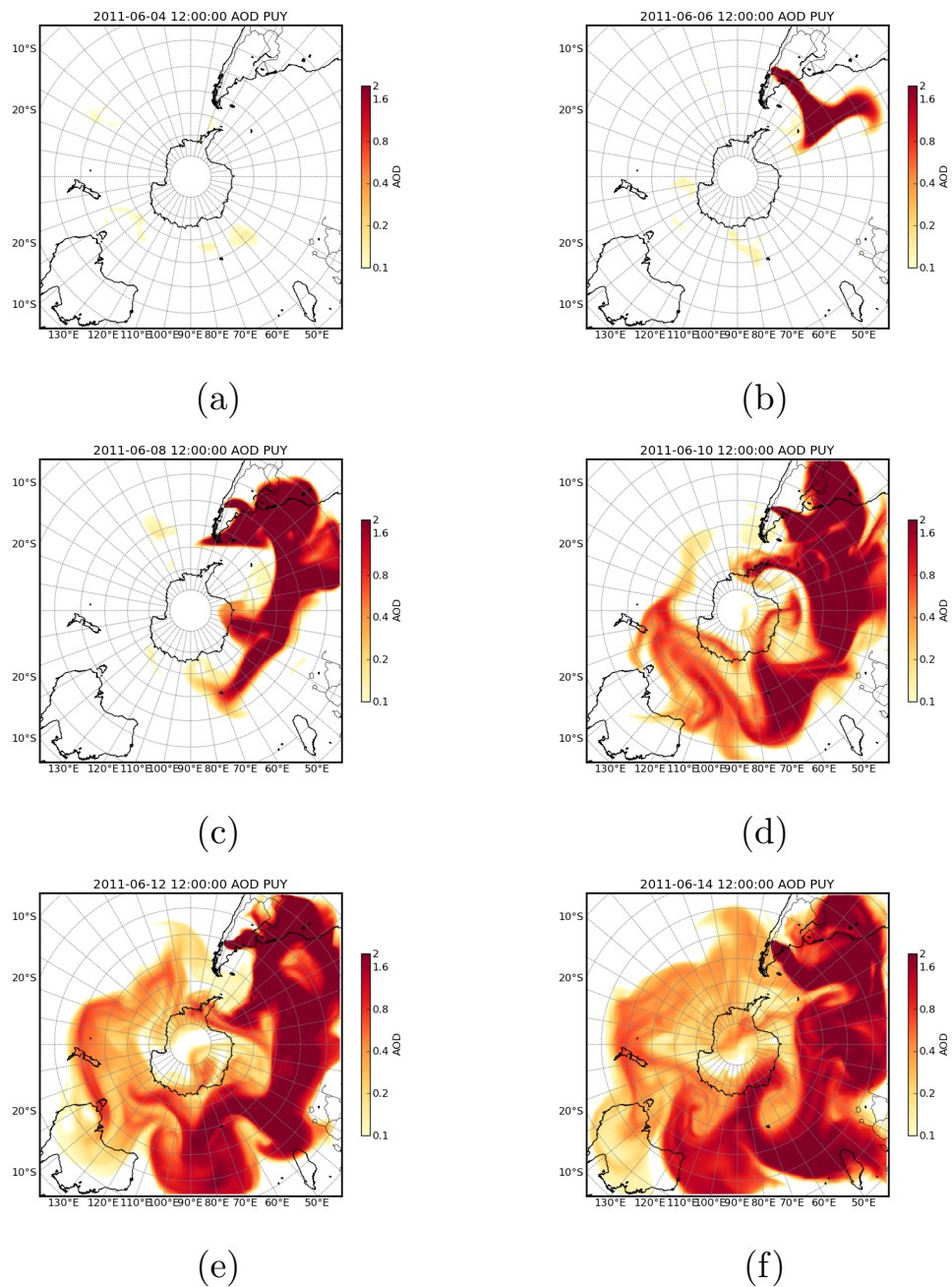

**Figure 9.** *Simulated AOD at 600 nm every 48 hours from June 4, 12UTC to June 14, 12:00 UTC*

| day | H | $\dot{M}$ | M |
|---|---|---|---|
| 04/06 - 19-24UTC | 10 | 250 | $5.21 \times 10^7$ |
| 05/06 - 00-08UTC | 10 | 250 | $8.33 \times 10^7$ |
| 05/06 - 08-20UTC | 10 | 110 | $5.50 \times 10^7$ |
| 05/06 - 20-24UTC | 10 | 60 | $1.00 \times 10^7$ |
| 06/06 - 00-19UTC | 10 | 60 | $6.00 \times 10^7$ |

**Table 8.** H : column height (km) ; $\dot{M}$ : Mass flux ($kt\,d^{-1}$); M : emitted mass of $SO_2$ (kg)

### 7.3    Analysis of the circumpolar transport

The simulation is initialized by climatological concentrations for aerosols and trace gases from the LMDZ-INCA Chemistry-transport model. These two datasets are also used to provide the top and lateral boundary conditions during the simulation. The simulation itself, covering from May 15 through June 30 can be divided into two successive phases : first, from May 14 to June 4, the model undergoes a spinup

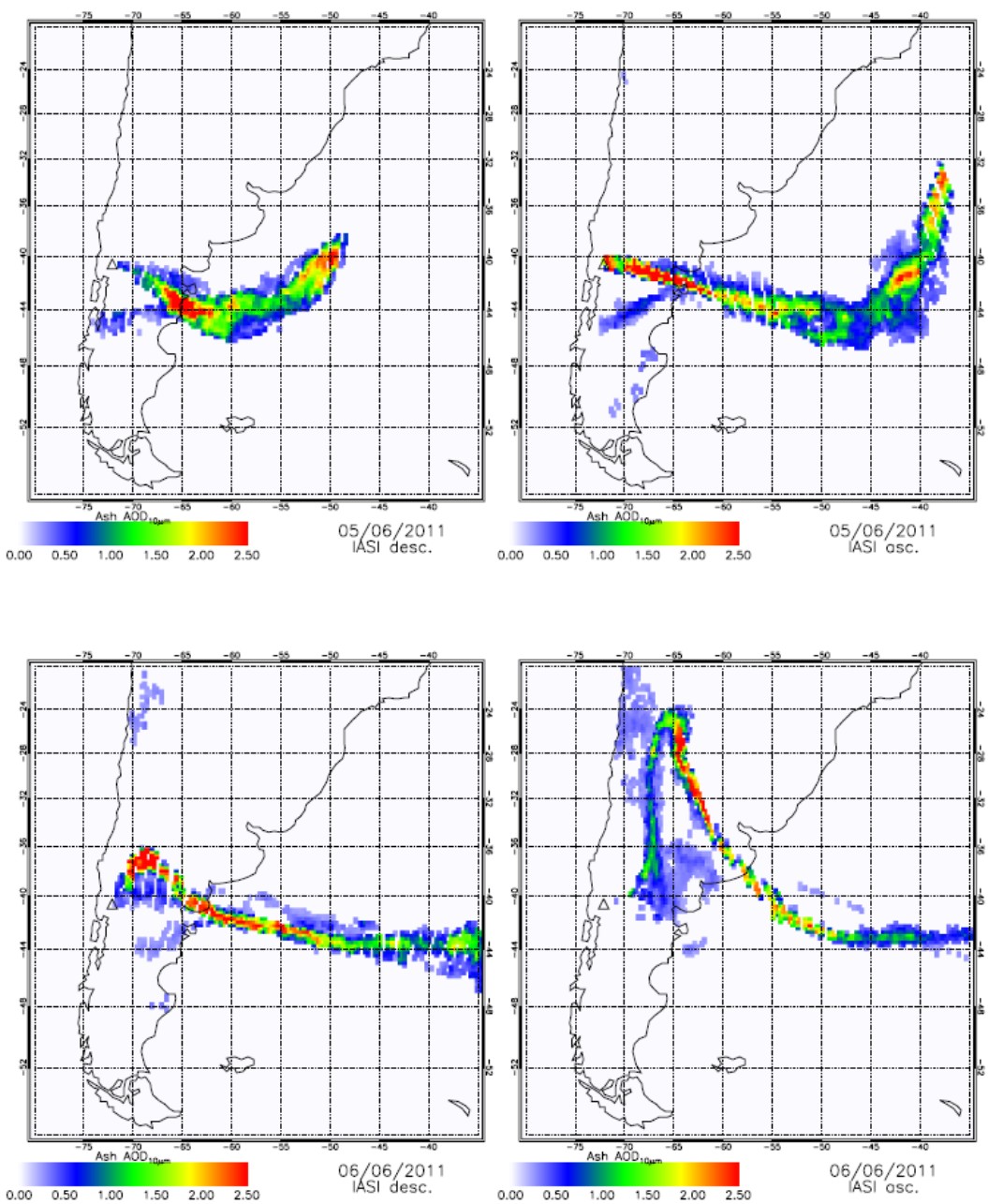

**Figure 10.** *Figure by L. Klüser, T. Ebersteder and J. Meyer-Arnek, published in Klüser et al. (2013) as Fig. 2, with the following description: "Ash Optical Depth at 10 μm of the PCCE plume for 5 through 6 June. Descending (desc.) orbits represent morning observations, ascending (asc.) orbits are from local evening. The black triangle indicates the position of the volcano."*

period, with the concentrations of gaseous and particulate species building up due to the emissions of sea-salt and anthropogenic contaminants (Fig. 9a). At the end of this spin-up period, significant AOD values, from 0.05 to 0.20 appear over the southern ocean from 30 to 70°S, mostly due to sea-salt emissions, consistent with the findings of Jaeglé et al. (2011), and consistent with the satellite-based climatology

of these authors, which represent a mean value about 0.15 in these areas. In the subsequent time steps, the volcanic ash plume from the Puyehue volcano becomes the dominant feature of the AOD structure in the southern hemisphere. While it is difficult to compare the simulated values to measured ones because of the large uncertainties on the mass flux and size distribution of the volcanic ashes, it is possible to com-

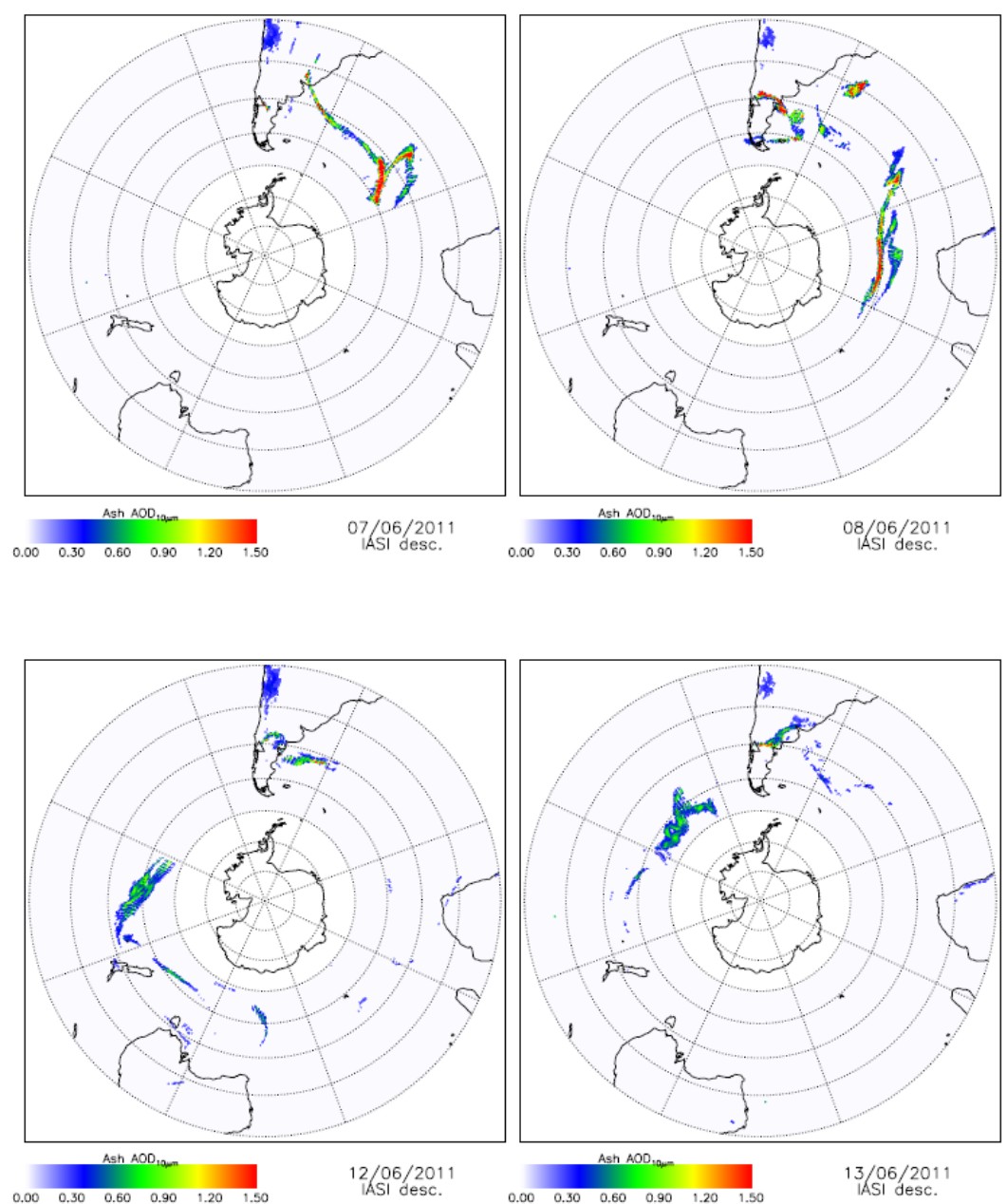

**Figure 11.** *Figure by L. Klüser, T. Ebersteder and J. Meyer-Arnek, published in Klüser et al. (2013) as Fig. 3, with the following description: "The PCCE ash plume on its way around the Southern Hemisphere for descending MetOp orbits from 7, 8, 12 and 13 June."*

pare the modelled trajectory of the ash plume with space-borne observations. For this purpose, we will rely on the space images and analyses provided by Klüser et al. (2013) and Global Volcanism Program (2013). Fig. 9b for June 6 at 12UTC (8AM local time) can be compared to Figure 2 of Klüser et al. (2013), reproduced here for the Reader's convenience as Fig. 10, which shows that at this time, about 36 hours after the onset of the eruption, the initial direction of the volcanic plume is eastward, with a slight southward

tilt, consistent with the CHIMERE simulations. On June 8 (Fig. 9d), the simulated pattern for ash transport also fits very well the pattern that is visible on Fig. 11 (also taken from Klüser et al. (2013)), with the initial portion of the ash plume travelling southward over the southern Atlantic and reaching towards the southern Pacific ocean over cape Horn, a pattern that is observed in both CHIMERE observations and the satellite observations. The older parts of the plume are located off the Atlantic coasts of Argentina, also covering a

large part of southern Brazil in the model but not so in the infrared AOD data (Fig. 11). Finally, the plume from the initial explosions are located at that time in the southern ocean, in-between the southern tip of the African continent and the Antarctic. It can also be observed that while the ash plume is continuous in the CHIMERE simulation, it is not so in the observations. This reflects the succession of explosive phases and quiet phases of the volcanic eruption, while the flux imposed to the CHIMERE model is continuous, as discussed in Boichu et al. (2013), who also present a possible workaround for this problem by assimilation of satellite data.

Four days later, on June 12, the leading edge of the volcanic ash plume is located at about 135°W and 55°S above the southern Pacific ocean, while other portions of the plume are located above New-Zealand, Tasmania, and areas of continental Australia and South-Africa (Figs. 9e and 11). Later on, on June 14, the leading edge of the ash plume reaches back to the southern coasts of Chile, as visible in both the simulation outputs (Fig. 9f) and the report of Global Volcanism Program (2013), which indicates that part of the plume was reaching South-America from the Pacific ocean at that time between 35 and 50°S while other parts of the ash plume were located further to the South, close to the Antarctic peninsula, consistent with Fig. 9f. On June 14 and during the following days, the plume from the initial explosion of June 4 and the following days is overpassing the Puyehue volcano again, a fact that is correctly captured by the CHIMERE model.

## 8    Conclusions

CHIMERE-2017 is a model version which presents several major improvements compared to the earlier version described in Menut et al. (2013a). Compared to the previous model version, anthropogenic emissions can be generated anywhere in the world from the HTAP emission inventory, as well as mineral dust emissions, which were available only for North Africa and the Arabian peninsula in previous model versions. With the same objective of permitting the use of the model in any part of the world and at any scale from urban to hemispheric scale, an important limitation of the model has been removed by improving the internal treatment of the transport on the sphere, allowing for domains up to the hemispheric scale, and possibly including a geographic pole. Much attention has also been paid to the physical processes, including a major update in the representation of the physical processes affecting the aerosols, as well as the effect of the modelled aerosol on the photolytic reaction rates. Other efforts have been made to improve the user's experience with the model: this includes improvements in the parallelization of the model in order to reduce computation time, as well as providing key observable variables such as the Aerosol optical depth and LIDAR backscatter coefficients, which permits

the user to compare the outputs of the model directly with the results of remote-sensing observations.

These improvements pave the way to many applications that were out of reach for the CHIMERE model up to now: CHIMERE 2017 has the necessary abilities to give new insights on questions such as the radiative impact of aerosols on photochemistry, at all scales, from urban to hemispheric, including mineral dust emissions and deposition anywhere in the world. The possibility to run hemispheric simulations also allows the use of this CTM for the study of transport of aerosol and gaseous contamination plumes between the different continent within a hemisphere. It contributes to bridge the gap between global chemistry-transport models such as LMDz-INCA, MOZART or Geos-CHEM and regional models: while CHIMERE has already been used successfully for the evaluation of the decadal trends in air quality over Europe (Colette et al., 2011), as shown by the study Xing et al. (2015) with the hemispheric version of CMAQ, hemispheric versions of regional CTMs are tools that can be used successfully to study long-term trends in regional air quality with added value from models simulated in regional domains only because they can perform a consistent simulation over the entire hemisphere without relying on boundary conditions provided by global CTMs relying on different assumptions and parameterizations.

## Code availability

The present article refers to the CHIMERE-2017 release, which is freely available and provided under the GNU general public licence[4]. The source code along with the corresponding technical documentation can be obtained from the CHIMERE web site at http://www.lmd.polytechnique.fr/chimere/.

*Acknowledgements.* For anthropogenic emissions, EuroglobalMap products includes Intellectual Property from European National Mapping and Cadastral Authorities and is licensed on behalf of these by EuroGeographics. Original product is freely available at www.eurogeographics.org. Terms of the licence available at http://www.eurogeographics.org/form/topographic-data-eurogeographics. The MACC boundary conditions data set was provided by the MACC-II project, which is funded through the European Union Framework 7 programme. It is based on the MACC-II reanalysis for atmospheric composition; full access to and more information about this data can be obtained through the MACC-II web site http://www.copernicus-atmosphere.eu.

We acknowledge C. Prigent for providing the global high resolution aeolian aerodynamic roughness length. The authors would also like to acknowledge L. Klüser, T. Ebersteder and J. Meyer-Arnek for publishing the figures here re-used as Figs. 10 and 11 with an Creative-Commons license, permitting reuse these figures. We are also endebted to the two anonymous Reviewers who helped improve a lot this study from a scientific and editorial point of view.

---

[4]http://www.gnu.org/copyleft/gpl.html

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
