# Peer review of "CHIMERE-2017: From urban to hemispheric chemistry-transport modeling"

_Geoscientific Model Development, 2016_

## Referee Comment (RC1) · Anonymous Referee #1 · 7 Nov 2016

This manuscript details the recent updates made to the CHIMERE chemical transport model to extend its capability to include urban and hemispheric modeling. As pollution concentrations have fallen in many areas in the past couple decades, the problem of air pollution has expanded from being primarily a local/regional issue to a more hemispheric/global issue. Therefore, having air quality models capable of simulating the hemispheric to local scale are becoming more important. This work attempts to detail the vast number of updates made to the CHIMERE modeling system in order create a more versatile modeling system. The authors and developers of the model should be commended on what was surely a very large undertaking to update almost every aspect of the modeling system. The manuscript also includes a section that attempts to evaluate the performance of the new modeling system, which in my opinion is the weakest section of the manuscript.

My general comment on the evaluation of the CHIMERE-2016 modeling system is that it really only provides model performance metrics for a very coarse modeling domain (0.5 X 0.5 degree). In addition, wherever the model performance is poor, the authors simply seem to blame the emissions for the poor performance. While emissions certainly can/do contribute to model performance issues, they are far from the sole cause of poor model performance. In my opinion, the evaluation provided does little to provide real faith that the new modeling system can accurately simulate air pollution across a multitude of scales. Also, the volcanic ash case study simulation provided seems only to establish that the model physics does not fail while traversing the southern hemisphere. No evaluation of the actually particle concentrations for the case study is really provided, so it's not clear whether the model can actually simulate the fate of the emitted ash particles (e.g. effects of particle transformation and deposition) with any real accuracy.

It would very useful to have a much more detailed evaluation of the modeling system, including simulations at various horizontal grid resolutions and domain sizes (e.g. urban at fine grid resolution, regional at moderate grid resolution, and hemispheric at coarse model resolution). This may be too much to add to this manuscript, so perhaps it would be better if the authors tackled this as a Part I/Part II series, with Part I describing the updates to the modeling system (as has been done), and Part II providing a detailed model evaluation of the system. I think this would end up being a much more useful set of papers to potential CHIMERE users in my opinion. As it stands, I don't feel like the evaluation in the manuscript provides enough support for the amount of work that was done in updating the model.

I've provided a number of specific comments in the PDF provided with this review.

Please also note the supplement to this comment:
http://www.geosci-model-dev-discuss.net/gmd-2016-196/gmd-2016-196-RC1-supplement.pdf

[Figure]

**Supplement:**

[revised manuscript text omitted]

---

## Referee Comment (RC2) · Anonymous Referee #2 · 16 Nov 2016

General Comments. The manuscript introduces the new version of the CHIMERE chemical transport model. The text is rather long and requires an English proofreading. Also, providing examples and evaluations with observational data of several new model features will make the paper further compelling. Besides the questions listed below, I recommend to be added to text: 1) A table of variables presented in text, 2) A table describing all physical input variables that are needed from a meteorological model (as WRF or IFS/ECMWF) to drive a simulation with this version of CHIMERE.

Questions/Comments Pag 2, abstract, lines 4-10 It seems that there is a mix-up between the words 'scales' and 'domain sizes' . The authors should refer to scale as the smallest eddy resolved by the model and not mix with the size of the computational grid used to simulated certain phenomena. In this sense, stating that CHIMERE-2016 can be applied at any scale seems to be unsuitable. I invite the authors to rephrase

those sentences.

Pag 2, Introduction State clearly the class of atmospheric-chemistry models which CHIMERE-2016 fits. Is it a coupled online atmospheric-meteorological model? If yes, does it include feedbacks between atmospheric composition and the model dynamics? How does it compare with other state-of-the-art models like WRF-Chem and COSMO-ART?

Pag 3, line 5 State clearly the means of 'CHIMERE core' . Did you say 'dynamic core'? Page 3, line 20. The expression master/slaves are more common in this context.

Page 4, lines 1 – 3. In the new version, clarify if the model output is split into several files (each slave writes its model solution in a particular file) or if each slave writes its own sub-domain but in a single file, which comprises the entire domain.

Page 7, Line 20 Include a brief description of the numerical properties of the advection scheme applied in this chimere version to transport scalars.

Page 8, section 3.1 The new version aims to simulate tracer transport on continental/hemispheric scales with the model top at 200hPa. How the organized vertical transport of pollutants associated with convective and moist plumes are handled in this configuration?

Page 16, section 5. Describe the numerical solver of the chemical mechanism applied in this model version.

Page 28, line 12 The emission configuration does not agree with the general observed 'umbrela' shape of the volcanic plume. Page 29, Section 7.4 The simulation outputs discussion lack comparison with observational data. It would be very instructive for the readers to perceive the fidelity of the model transport and AOD simulations.

Page 32, Conclusions. Line 16: Is it true that this version 'has the ability to include all types of emissions' ?

---

## Author Comment (AC1) · 15 Mar 2017

**CHIMERE-2017 : from urban to hemispheric chemistry-transport modeling**
by S. Mailler et al.

Answer to Anonymous Reviewer 1

This manuscript details the recent updates made to the CHIMERE chemical transport model to extend its capability to include urban and hemispheric modeling. As pollution concentrations have fallen in many areas in the past couple decades, the problem of air pollution has expanded from being primarily a local/regional issue to a more hemispheric/global issue. Therefore, having air quality models capable of simulating the hemispheric to local scale are becoming more important. This work attempts to detail the vast number of updates made to the CHIMERE modeling system in order create a more versatile modeling system. The authors and developers of the model should be commended on what was surely a very large undertaking to update almost every aspect of the modeling system. The manuscript also includes a section that attempts to evaluate the performance of the new modeling system, which in my opinion is the weakest section of the manuscript.

First of all, we would like to thank Reviewer 1 for his/her careful reading of our paper and many useful comments, and evaluating in a careful and fair way the strengths of the manuscript and its limitations.

Please note that the title of the paper has been updated to CHIMERE-2017 instead of CHIMERE-2016 since a new model version has been issued, with only bugfixes from the 2016 version. All the simulations have been redone with this corrected version, which does not bring any change in the discussion since the changes are only small bugfixes.

Below are, in blue, the comments and requests by Reviewer 1, in black our answers, and in green the description of the corresponding changes that have been brought to the Manuscript, when relevant.

My general comment on the evaluation of the CHIMERE-2016 modeling system is that it really only provides model performance metrics for a very coarse modeling domain (0.5 X 0.5 degree).

Yes, this is true. However, we refer more explicitly in the last version of the paper to recent studies (e.g. Markakis et al., 2015, Terrenoire et al. 2015) giving thorough validations of the model for regional/urban scale, including an article by Bessagnet (2016) describing results from the multimodel comparison EURODELTA III project, which gives detailed information about model performance and comparison to other CTMs.

This is done in the introduction, p. 2, l. 25-49

In addition, wherever the model performance is poor, the authors simply seem to blame the emissions for the poor performance.

This maybe seems like that but, for example regarding SO2, what we actually blame for poor performance is not the emission inventory itself, but rather the vertical disaggregation profiles, which are model developers' choice (our choice) but are highly uncertain, and assume that vertical emission

profiles are uniform in time independantly of meteorology etc. Which is of course questionable. This is now mentioned explicitly as a model limitation (p. 18, l. 22-34)

For NOx emission, we now refer more precisely to Terrenoire (2015) (p. 18, l. 13-21) who makes a more precise statement about possible missing emissions, but of course we are aware that emissions are far from being the only source of error in simulated concentrations, and we hope that this appears in the manuscript.

In my opinion, the evaluation provided does little to provide real faith that the new modeling system can accurately simulate air pollution across a multitude of scales. Also, the volcanic ash case study simulation provided seems only to establish that the model physics does not fail while traversing the southern hemisphere. No evaluation of the actually particle concentrations for the case study is really provided, so it's not clear whether the model can actually simulate the fate of the emitted ash particles (e.g. effects of particle transformation and deposition) with any real accuracy.

This is correct, but actually, when changing the model structure from lat-lon to other coordinate system allowing modelling of circumpolar transport, the most challenging part was to actually obtain a model that was able to perform the transport without errors, because the rewriting of the transport schemes was using a very different discretization principle than in previous model version, so this simulation is aimed at showing that the model is able to have the volcanic plume arrive at the right time at given places where it has been actually observed. This is what we intend to show by comparing our results to published reports about the chronology of the Puyehue ash plume in a variety of locations.

Simulating the right concentrations for ash and the right AOD is more dependant on the model forcing by emitted volumes of ash and their size distribution, which are so uncertain that modellers can just adjust them by trial and error to obtain the desired concentrations and AODs (for example, the $m_{63}$ parameter that permits to evaluate the fraction fo fine ash / total ash can vary from about 0.01 to about 0.6 from one eruption to another and it is very difficult to have reliable measurements of this parameter). For these reasons, we chose to just have a first estimate of the emissions according to published data/methods, and focus on the chronology, because the part of the model that was modified most deeply in updating for regional to hemispheric version is the transport, so this is what actually needed to be checked, and is in our opinion verified since the modelled plume is back to the Chilean coast and at the correct location on June 13/14 as observed.

However, we hope to obtain better estimates of ash and SO2 emissions by working with specialists of volcanic emissions and perform a comparison of AOD and SO2 columns in a further study.

It would very useful to have a much more detailed evaluation of the modeling system, including simulations at various horizontal grid resolutions and domain sizes (e.g. urban at fine grid resolution, regional at moderate grid resolution, and hemispheric at coarse model resolution). This may be too much to add to this manuscript, so perhaps it would be better if the authors tackled this as a Part I/Part II series, with Part I describing the updates to the modeling system (as has been done), and Part II providing a detailed model evaluation of the system. I think this would end up being a much more useful set of papers to potential CHIMERE users in my opinion. As it stands, I don't feel like the evaluation in the manuscript provides enough support for the amount of work that was done in updating the model.

In our view, this manuscript is a model description manuscript more than model evaluation, since a

number of studies already use the CHIMERE model and compare it to observations and to other comparable models, giving to the interested users a detailes view of the performance of the model depending on the species, the resolution, etc. We added some information in this sense in the revised version, including more references to detailed evaluation papers such as Terrenoire (2015) or Bessagnet (2016) . Even though these papers use earlier versions of the model, the model performance in its "classical" regional configuration only evolves smoothly with time even when lots of new developments are included, this is also the case between the described version and earlier versions that have been thoroughly evaluated. Therefore, the focus in this paper is not put on evaluating a strong increase in model performance, which is not the case (the evolution of modelling scores from one version to the next is slow and smooth), but a change in the scope of the model, including more processes, and extending the capabilities in terms of domain size to the hemispheric scale. This precision of the scope of the manuscript can be found in the introduction of the revised paper, p. 2, l. 29-46.

I've provided a number of specific comments in the PDF provided with this review.

Thank you very much, these specific comments have been adressed directly when it was typos or propositions for rephrasing (thank you very much for signaling them), for more substantial comments you will find the answers below.

p. 2, l. 14 : Missing reference for MELCHIOR chemical scheme

The original reference in to the PhD thesis of Mireille Lattuati :
Lattuati, M.: Contribution à l'étude du bilan de l'ozone troposphérique à l'interface de l'Europe et de l'Atlantique Nord: modélisation lagrangienne et mesures en altitude, Phd thesis, Université Pierre et Marie Curie, Paris, France, 1997

However, the model is extensively described with lists of species and reactions in a more recent paper by Menut et al. (2013), which is now cited, easier to find and in English (cited for that purpose on p. 2, l. 70).

p. 3, l. 13-15 : Important to note here that in the current offline configuration no radiative feedback effects to the meteorology are available, which can be important when simulating aerosol effects, particularly on the global scale.

The following sentence has been added : "Another important point is that even though the processing of meteorological input has been changed as described here, the present version offers does not take into account any radiative or microphysical feedback of atmospheric chemistry on meteorology." (p. 2, l. 76-79)

p. 3, l. 26 : It would be useful to clarify exactly what is meant by "workers". In this case, the processors doing the grid computations.

This has been done :by replacing "workers" by "the various slave processes that performed the actual gridded calculations" (replacing "workers" by "slaves" is a requirement of Reviewer 2)

p. 4, l. 3-6 : This would seem an appropriate place to discuss the runtimes for some different model configurations (e.g. regional vs. global with varying horizontal and vertical grid spacings). I don't recall

the model runtime being discussed in detail anywhere else in the manuscript.

While we agree that it would be interesting to provide such a discussion, we did not perform the necessary simulations and computational tests to do it, which would involved a controlled computational environment – particularly regarding i/o processes that may be competing with the test simulations, tests with different resolutions, different numbers of cores, etc.

p. 8, l. 14-15 : This is very low model top for a model simulation, particularly at the continental/hemispheric scale. There needs to be some discussion of how the boundary at the top of the model is handled, and that in the case of such a low model top, the model providing the boundary conditions extremely important to accurately simulating on such a large scale.

The statement in l. 14-15 is as follows : "The pressure of the top of the model, $p_{top}$, typically from 500 hPa for studies at urban/regional scales to 200 hPa for continental/hemispheric scale studies."

While these pressure levels of model top would be very low for a meteorological simulation, making the choice of the boundary conditions critical, we have the feeling that for chemistry-transport simulations this is not so because most trace gases have either very small or slowly variable concentrations at this altitude, with the notable exception of ozone. Therefore, the boundary conditions at the model top and boundaries are taken from climatologies, typically from LMDZ-INCA simulations for all species except for mineral dust for which climatological concentrations are taken from GOCART.
Note that the model top is an adjustable parameter and has only the meteorological driver as constrainst.
In our case and in this revised version, the hemispheric simulation was actually performed with a model top of 100hPa. The sentence with 'typically' was changed and is now:  "The pressure of the top of the model, $p_{top}$, can be freely set by the user with typical values from 500 hPa for studies at urban/regional scales to 100 hPa for continental/hemispheric scale studies." (p. 6, l. 49-52)

p. 10, l. 12-17 : So the user is able to define their own splits for these species if they wish?

These speciation tables are described in the model documentation, and the users may modify them to take into account their own splits if they wish top use other splits than described here.

p. 10, l. 20-22 : Is there a lower limit to a user specified layer one model height? What if a user specifies a very low layer one height?

Actually the statement was erroneous : Residential emissions are affected not to the first model layer but to a layer between 0 and 20m height, which might overlap on the second model layer if the first model layer is very thin. In the revised version, we provide a reference to Terrenoire et al. (2015) where the detailed profiles that are used for vertical disaggregation are presented. (p. 7, l. 53-57)

Regarding the precise question the Reviewer is asking, there is no lower limit to the user-specified thickness of the first layer model. As specified p. 6, l. 42-46, this parameter is freely set by the user. It is the user responsability to choose values such that the model configuration is reasonable.

p. 10, l. 23 : How exactly is the layer in which these emissions are emitted determined? Is any plume-rise calculation still being performed? It's not clear from the text exactly what method is being used to determine the layer the emissions are emitted.

No plume-rise is applied, fixed emission profiles are used for each SNAP sector, as described in Terrenoire (2015). The following sentence has been added :

The vertical distribution profiles that are used for each SNAP (``Selected Nomenclature for Air Pollutants") sector are constant profiles depending only on the SNAP sector, and are presented in Terrenoire (2015) (p. 7, l. 53-57).

The absence of a plume-rise representation is now explicitly mentioned p. 18, l. 30-34 as a possible reason for the lack for too weak SO2 concentrations.

p. 15, l. 11 : Why is that? Does resuspension not also occur over non-urbanized surface? Please explain the rational for only applying the resuspension over urbanized surfaces.

The resuspension process differs from the emissions. In the model, we consider only the traffic-related resuspension from the traffic-related emissions. This is why this source is taken into account only in the PPM species (anthropogenic) and in urbanized areas. But we agree with the Reviewer than resuspension process may occur in other environment. This is just not taken into account for the moment in this model version. The title of this paragraph has been changed to "traffic-related resuspension", making it clear that not all resuspension processes are taken into account.

p. 17, l. 9 : What specifically are "various environments"? Please give at least several examples here.

The following sentence has been added (p. 12, l. 21-25):

If the role of halogen chemistry was traditionally considered limited to the marine boundary layer, recent observations have shown significant $ClNO_2$ concentrations from few ppt in Mid-Continental Urban Environment (Mielke, 2011) to 2000 ppt in the Coastal Marine Boundary Layer (Riedel, 2012).

p. 18, l. 10-11 : How reasonable is this assumption? Is there a reference available? Or is it stated in Nenes et al.?

Yes, ISORROPIA is a model for inorganic aerosols only, and its original version described in Nenes et al. (1998) makes the hypothesis that mineral dust and organic aerosols are not hygroscopic. Even though ISORROPIA II (Nenes et al. 2007) takes into account the solubility of some components of mineral dust, this is not taken into account into CHIMERE, partly because CHIMERE so far does not include information on the composition of mineral dust.

p. 19, l. 10 : Already referenced and explained in the previous section.

This has been corrected, thanks.

p. 21, l. 5 : Please elaborate on what condensed phases of water (e.g. ice particles, snowflakes, graupel, etc.) are used in the calculation of extinction from cloud water and ice. How are the optical properties determined from cloud water and ice?

Following the recommandations of the Fast-JX developers, the effective size of ice particles is estimated following Heymsfield (2003), J. Applied. Meteorol. :

REFFI = 164. * (IWC**0.23)

Where Reff,i is the effective radius of ice particles, and IWC is the ice content of the atmospheric particles (g/m3).

Regarding water droplets, their radius is estimated also following the recommandations of fast-JX developers, as 9.60 µm for clouds at low altitudes (below 810 hPa), 12.68µm for high clouds (above 610 hPa), and linearly interpolated between these two values for intermediate altitudes. We are not aware where these recommendations come from, however, they are consistent with the typical radius of cloud water droplets.

This is now mentioned in detail in the article (. 14, l. 36-44).

p. 21, l. 11-12 : Does light extinction from clouds include cloud overlap or overlay? If so, how?

No. Light extinction for clouds unfortunately does not take into account the possibility of fractional cloud covers and the possible overlaps between successive cloud layers. The cloud fraction is not read from the meteorological model, and the cloud layers are assumed to be horizontally uniform in each grid cell. However, the succession of various cloud layers over the vertical dimension is taken into account in a complete way by the Fast-J radiative module, including the reflexion and diffusion of upward and downward fluxes by each cloud layer.

p. 23, l. 3 : Clarify what mixing model is used to calculate the optical properties of the aerosol size bins and why. The paper may give conflicting information. Section 5.3.3 states that an external mixing model is used but section 5.2.2 states that an internal mixing model is used for the wet aerosol diameter and density. If internal model determines the optical properties, give more details about the type of internal model such as volume weight average, core-shell, etc.

For optical properties, external mixing is assumed, while for other processes (thermodynamics etc.), the model assumes internal mixing. It is true that this choice can be surprising. It is well-known that the assumptions on the mixing state of aerosols can have an impact of their radiative properties, but in the CHIMERE model, so far external mixing is assumed, and each species interacts with solar radiation according to its own refractive index, and the corresponding terms of the Legendre expansion for its phase function.

p. 23, l. 10 : Please provide a reference for the ADIENT project.

The following sentence has been added :
The refractive indices reproduced in Table 5 are the ones provided along with the model, essentially based on the values compiled in the framework of the ADIENT project (http://www.reading.ac.uk/adient/refractiveindices.html, visited Jan. 17, 2017, as described by the corresponding technical report by E. J. Highwood (www.reading.ac.uk/adient/REFINDS/Techreportjul09.doc}, visited Jan. 17, 2017) – p. 15, l. 34-38

**Table 4 : Please state why Table 4 assigns OCAR refractive indicies associated with Brown Carbon. What type of organic material does OCAR represent?**

As for other species, we relied on the data provided by the ADIENT project and the corresponding technical report which gives the following explanations about Organic carbon :

"This discussion applies to the component of organic aerosol that is not black carbon and generally derives from fossil fuel burning. Since

the composition of this aerosol depends on combustion type and material it is problematic to prescribe parameters suitable for all occasions. Atmospheric organic crabon is often described as HULIS – Humic-Like Substances.  HULIS samples have been recorded with refractive indices of 1.595-0.049i (Pollution, Dinar et al, 2007), 1.622-0.048i (wood burning smoke, Dinar et al 2007), 1.56-0.003i (rural continental, Dinar et al, 2007), 1.65-0.0019i (biomass burning, Hoffer et al, 2006), 1.45-0.001i (Kreckov et al, 1993 – used in ADRIEX), 1.53-0.0055, (Koepke et al, 1997, used for POM in ECHAM). Measurements of the absorption properties of atmospheric HULIS use a variety of techniques to extract the HULIS from the aerosol. Some of these techniques can enhance the hydrophilic compounds for pollution and smoke samples. Perhaps for this reason, Swannee River Fulvic Acid (SRFA) has been used as proxy for atmospheric HULIS in numerous studies (e.g. Dinar et al, 2007) as it is available for laboratory measurements although some caution must be attached. Laboratory SRFA has a density of 1.47 g cm-3 and a refractive index of 1.634-0.021i at 532nm. Other groups use information from observations of biomass or fossil fuel burning to constrain their optical properties. For example, HADGEM2, the Met Office climate model, has separate optical properties for FFOC (fossil fuel organic carbon), SOA (secondary organic aerosol), and fresh and aged biomass aerosol. The real part of the refractive index for aged OC in HadGEM2 is taken to be the same as that for Biomass Burning (which is constrained by measurements at some wavelengths – 1.54 at 550nm) and the imaginary part of the refractive index is assumed to be wavelength independent at -0.006 (compared to that for aged biomass burning of 0.02 at 550nm). This value for OC is at the low end of the estimates for absorption from HULIS measured in the atmosphere and also lower than the absorption from SRFA.
It is clear that there is a range of uncertainty in the imaginary part of the refractive index that is appropriate for organic carbon. Even if one accepts that HULIS is appropriate, then this varies according to source and location. The aging process is also a complicating factor. The situation becomes more difficult if values at wavelengths other than 550nm are required. Kirchstetter et al (2004) report that the spectral dependence of light absorbing aerosols is dependent on the temperature and completeness of combustion – with high temperature combustion processes (such as diesel burning) exhibiting much less spectral dependence of absorption. They suggest a wavelength dependence of the mass absorption efficiency according to $\sigma = K\lambda\text{-}\alpha$ where $\alpha$ is the absorption angstrom exponent and $\alpha$=2.5 for biomass aerosols but only 1 for motor vehicle emissions of light absorbing aerosol. Kirchstetter et al (2004) also provide values of the imaginary part of OC refractive index between 350 and 700nm.
No single study or sample provides values for real, imaginary and wavelength dependent refractive index for OC. The recommendation made here is that SRFA refractive indices at 532nm be used as the anchor point for a wavelength dependent refractive index that retains the dependence of Kirchstetter et al (2004) between 350 and 700nm for the absorption part, but is wavelength independent in the real part (there being a lack of information regarding the wavelength dependence above 532nm) for the visible part of the spectrum. The value at 550nm is more absorbing than that used in HadGEM2, but still substantially less absorbing than HULIS isolated from several samples from different combustion and aerosol conditions.  For longer wavelengths still (above 4 micron), Hess et al (1998) wavelength dependence for WASO (water soluble type) is used as in Stier et al (2007). Extreme caution should be attached to the values at longer wavelengths.

Maybe these assumptions need to be updated or questioned, and maybe we should adopt other values for Organic Carbon. However, as mentioned in the manuscript, " the specification of these parameters is in a parameter file, and can be changed by the user to other values. In the same way, the user can easily introduce more species in the optical treatment for specific studies, e.g. volcanic ashes.", so it is easy for a user who wants to test the impact of other assumptions to specify other values of the refractive indices.

Fig. 7 : Presumably, the numbers on the right axis of each plot represent the mean value of MFB, MFE and PCOR and therefore should be the same as the read dot in the plot which also represents the mean of each statistic. This does appear to be the case for MFB and MFE, however there seems to be a large discrepancy between the PCOR values listed on the axis and those represented by the red dots on the plots. For instance, the PCOR for O3 summer is listed as 0.69 on the axis but the red dot lies on the 0.6 line. Can you please explain the discrepancy between the two values?

These numbers were actually not explained in the first version of the manuscript. They actually correspond to the overall value of the parameter, e.g. grouping in a single vector all the concentration values for all stations and all time points to obtain a spatiotemporal correlation (respectively, spatiotemporal MFB, etc.). the following sentence has been added to the figure caption :
"the values on the right axis of each panel are the overall value of the considered indicator, i.e. merging all the stations into a single statistical dataset as described in Jacobson (2005)".

P. 26, l. 3-5 : It could also be argued that 0.5 degree horizontal resolution is insufficient to simulate

urban environments that have a lot of heterogeneity in emission sources. How many of the remaining sites would be classified as urban and rural sites?

It is true that this resolution is not adequate to simulate urban (or industrial) environments. This is why we used EMEP stations for comparison, which are rural background sites. For example, in the reference EMEP publication by Thorseth et al., 2012, *Atmos. Chem. Phys.* States that "The EMEP monitoring sites are located such that significant local influences (local emission sources, local sinks, topographic features, etc.) are minimised. The basic idea is that the data should be representative for a larger region.".
The following sentence has been added : "Stations from the EMEP monitoring sites have been chosen for this study because their location has been selected in order to minimize local influences and be representative of large areas (Thorseth et al., 2012)". (p. 17, l. 17-20)

p. 26, l. 9-11 : Are these statistics for ozone for all hours of the day?

The statistics are for daily averaged values, which was not specified in the first version but is now specified (in the Figure caption and in the text).

p. 26, l. 15-16 : I highly agree with this statement and needs to be addressed more prominently in the manuscript in my opinion. It should be made clear to the reader that the performance of the model at this resolution for anything but background stations should be considered with caution. I would expect considerably different model performance when the resolution of the simulation is increased to a more regional/urban scale.

(I assume that the Reviewer's remark refers to lines 18-19 which are a statement about model's resolution and its possible impact on scores for nitrogen oxydes).
As stated above (in the revised version), the EMEP stations are designed to be representative at least at a regional level, which normally reduces the impact of local factors such as industries or major urban areas and therefore permits comparison with a simulation at 0.5° resolution. However, of course, this is never totally the case and local influences, even though they may be small, can sometimes be present, but by using the EMEP stations that are expert-selected to be representative at regional level we hope to reduce the effect of insufficient model resolution. Detailed reference to studies with higher resolutions is now provided in the Introduction, taken from large international multimodel comparison projects such as AQMEII, Eurodelta (I and III) as well as from urban-scale studies (p. 2, l. 3-49)

p. 26, l. 19 : What emissions specifically are they saying are underestimated?

Unfortunately, the statement in the conclusions of Terrenoire 2015 is not very explicit about this underestimation. The exact statement made in this paper is as follows : "The difficulty for the model in reproducing NO2 concentration is likely to be due to the general underestimation of Nox emissions, especially during the traffic daily peaks, as well as a horizontal resolution that is not high enough to represent correctly the spatial gradients of the emissions over medium and small cities.". The second part of this statement only partly applies to the present study since for the present study only EMEP background measurement sites are retained. We added the words "especially during the traffic daily peaks" to the revised version to make the statement a bit more explicit.

p. 26, l. 22-24 : Can the authors say something about the SO4 performance of the model? Is SO4 underestimated, suggesting issues with the conversion of SO2 to SO4 in the model? It's easy to simply blame the emissions, but more information is needed on the overall performance of the model before

simply blaming emissions for the poor SO2 performance. And what exactly is the issue with the vertical distribution of SO2? Is this suggesting not enough SO2 is being elevated to upper layers in the model? Presumably the primary SO2 sources are energy generating units, which typically emit above layer one. Is this a plume rise issue in the model?

p. 27, l. 1-4 : What about the performance of SO4?

p. 27, l. 5-10 : So the model is overestimating both SO4 and SO2 in the winter? Is SO4 also overestimated in the summer?
(We group the answer to these three remarks)

We agree that it is easy and occasionnaly convenient to simply blame the emissions, but is was not our intention here, since we point as a possible error source the vertical profiles for anthropogenic emissions, which are a choice of the model developers, described in Mailler et al. (2013) and Terrenoire et al. (2015). So, what is pointed as a possible error source here is actually a fallback of the model which uses fixed vertical emission profiles (this is now mentioned explicitly, p. 18, l. 30-34). If this is the case, the errors due to this modelling choice would be reflected essentially on SO2 concentrations since SO2 is the species that is the most influenced by industrial stack emissions. We also give more general information on the model's characteristics regarding SO2 ans sulphate, we added more details and references to this part :

"The SO2 shows the largest MFE for both summer (74.5 %) and winter (80.2 %) and quite low correlation in summer (0.20). It shows positive bias in winter: MFB=35.5%. The difficulty of SO23 simulation could be related to the uncertainties in the vertical profiles on emissions, which is a particularly sensitive factor in SO2 simulations since industrial stack emissions represent a substantial part of SO2 emissions (Pirovano et al. 2012, Mailler et al. 2013). While other CTMs have included a plume-in-grid model for subgrid treatment of point emissions depending on the actual meteorological conditions and flux characteristics, this is not the case of the CHIMERE model, which can also limit the performance of the model regarding SO2 concentrations. The conversion of SO2 to sulphate can also be a source of error on SO2 concentrations, as mentioned by Ciarelli (2016) and Bessagnet (2016) who observed very different behaviour of models far from emission sources probably due to the chemical mechanisms. The lower correlation coefficient in summertime  is present in all the CTMs examined in Bessagnet (2016)." (p. 18, l. 30-41)

p. 27, l. 5-10 : Are there soil measurement available that could be used to indicate whether or not dust is a major contributor to the bias?

We agree that this statement : "*This might indicate that the dust, whose emissions are very sensitive to the wind speed, contribute to the PM$_{10}$ errors in winter*" is probably too speculative, and we are not aware of routine measurements of mineral dust concentrations that would permit to readily verify this (at least on the EMEP measurement network only nitrate, sulphate and ammonia are measured, to our knowledge). So we removed this statement.

p. 27, l. 9 : These are not actually EPA guidelines. Boylan and Russell published this work due to a lack of clear guidance from the USEPA. Just to be clear.

Thank you for this useful precision. The notion of "EPA guidelines" has therefore been removed from

the text.

p. 27, l. 28 : Important to note that his is well below the stratosphere for most latitudes, so the specification of the the upper boundary conditions becomes very important. What values (i.e. model) was used to provide these boundary values? How well did that model simulate the volcanic plume?

This statement depends on what is meant by "most latitudes". Actually, if one refers to studies such as Seidel & Randel (2006, J. G. R., their Fig. 3) or Hoinka (1998, M. W. R.), the tropopause will be around 200hPa at 40°S in annual average, higher than that at the low (tropical) latitudes, lower than that at the polar latitudes, reaching ~250-300 hPa. At the latitude at which the plume travels, around 40°S, the top of model is about the same altitude as the tropopause.

However, for the reasons mentioned by the Reviewer, and also to avoid possible "leaks" of particles through model top, the simulation has been redone with model top at 100 hPa, which is higher than both the tropopause and the ash plume (so that the boundary condition at top of model does not need to include the volcanic plume).

As mentioned earlier in the article, the boundary conditions are from a LMDZ-INCA4 climatology.

p. 28, l. 13-15 : What exactly is occurring at the top of the model? This seems to be a very critical component of the global modeling that was performed that was not discussed at all.

Actually, at the top of the model, the concentrations are taken from LMDZ-INCA4 climatology, as for boundary conditions. The boundary condition at top of model is significant essentially for ozone concentrations, which are much stronger in the stratosphere and high troposphere than in the boundary layer. For other trace gases and aerosols, concentrations are very low in the stratosphere compared to their tropospheric concentrations.

p. 29, l. 3-4 : at least to eliminate initial conditions artifacts. What initial conditions were used?

The initial conditions have been taken from a climatology produced by the LMDZ-INCA model, as well as the boundary and top conditions. But this is only for the first time-step of the whole simulation. And to avoid the impact of this uncertainty, all simulations have at least two weeks of spin-up before analysis of the results.

p. 29, l. 2 : My general comment on this analysis is that it represents an evaluation of the meteorology and not the chemistry in CHIMERE. If the simple point of this analysis is that the model can transport a plume across a hemisphere is an at least reasonable fashion, than this analysis is probably sufficient, and should be stated as such. However, the analysis does not provide much faith that CHIMERE can accurately simulate the plume in terms of concentration and composition, which would be a much more useful evaluation.

Fig. 8 : It would be useful to include plots of observed AOD as well for comparison.

It is true that comparison to observational data is missing. However, making a comparison with observational data would require a better input regarding volcanic emissions, which we are not able to do at this point. As an example, the parameter $m_{63}$ quantifying the proportion of fine ash / total ash, is highly uncertain, as well as the total ash quantities emitted, etc. Therefore, we would prefer not to

present comparison to AOD data before this problem is solved, and stay with a qualitative discussion relying on published data analyzing observations to check that the chronology of the plume transport is represented in a realistic way in our simulation.

p. 32, l. 5-11 : I don't find this section to be very enlightening without some comparison to observed value. If the goal is simply to imply that the lidar plots seem to produce reasonable patterns, than it should be made very clear that that is the only goal of creating the plots.

We agree with the Reviewer that this section does not bring much to the study, given that the methodology for calculation of the LIDAR observables is already described earlier, and this plot lacks comparison with actual LIDAR data. Therefore, this subsection has been removed, as we cannot really challenge the idea that it is not very enlightening.

p. 32, l. 17-18 : Based on the limited simulations and analysis performed here, it's difficult to say whether the model can indeed be used to simulate urban to hemispheric scales with reasonable accuracy. No urban or regional scale simulation is presented here.

The reason why urban scale simulations are not presented here is because simulated aurban scale processes is already an existing feature of CHIMERE (as described in the studies of Markakis et al (2015), Valari et al. (2008, 2010) largely cited and described in the present study (p. 2, l. 3-25), while simulating hemispheric scale transport is one of the most significant new features of this version compared to the previous version that was described in Geophys. Model Dev. In 2013. Therefore, the focus is put on this new capability.

---

## Author Comment (AC2) · 15 Mar 2017

**CHIMERE-2017 : from urban to hemispheric chemistry-transport modeling**

by S. Mailler et al.

Answer to Anonymous Reviewer 2

Please note that the title of the paper has been updated to CHIMERE-2017 instead of CHIMERE-2016 since a new model version has been issued, with only bugfixes from the 2016 version. All the simulations have been redone with this corrected version, which does not bring any change in the discussion since the changes are only small bugfixes.

We would like to thank Reviewer 1 for accepting to review this paper, and for his/her useful remarks.

Below are, in blue, the comments and requests by Reviewer 1, in black our answers, and in green the description of the corresponding changes that have been brought to the Manuscript, when relevant.

**General Comments.**

The manuscript introduces the new version of the CHIMERE chemical transport model. The text is rather long and requires an English proofreading.

OK, we have performed a proof-reading of the article to try improving the language level and smooth out the differences between the style of the different coauthors. Also, reviewer 1 provided numerous improvements in the quality of language, which were very useful.

Also, providing examples and evaluations with observational data of several new model features will make the paper further compelling.

In our view, this manuscript is a model description manuscript more than model evaluation, since a number of studies already use the CHIMERE model and compare it to observations and to other comparable models, giving to the interested users a detailes view of the performance of the model depending on the species, the resolution, etc. We added some information in this sense in the revised version, including more references to detailed evaluation papers such as Terrenoire (2015) or Bessagnet (2016) . Even though these papers use earlier versions of the model, the model performance in its "classical" regional configuration only evolves smoothly with time even when lots of new developments are included, this is also the case between the described version and earlier versions that have been thoroughly evaluated. Therefore, the focus in this paper is not put on evaluating a strong increase in model performance, which is not the case (the evolution of modelling scores from one version to the next is slow and smooth), but a change in the scope of the model, including more processes, and extending the capabilities in terms of domain size to the hemispheric scale. This precision of the scope of the manuscript can be found in the introduction of the revised paper, p. 2, l. 29-46.

Besides the questions listed below, I recommend to be added to text:
1) A table of variables presented in text,
2) A table describing all physical input variables that are needed from a meteorological model (as WRF or IFS/ECMWF) to drive a simulation with this version of CHIMERE.

For the CHIMERE model, all variables are already cited and described in [Menut et al., 2013].
A table of all the variables that must/can be read from the Meteorological model is now provided in the

revised version (Table 1).

**Questions/Comments**

Pag 2, abstract, lines 4-10 It seems that there is a mix-up be-tween the words 'scales' and 'domain sizes' . The authors should refer to scale as the smallest eddy resolved by the model and not mix with the size of the computational grid used to simulated certain phenomena. In this sense, stating that CHIMERE-2016 can be applied at any scale seems to be unsuitable. I invite the authors to rephrase those sentences.

There is maybe a problem of vocabulary in different communities here. For the word "scale" as understood by the Reviewer, we would tend to use "resolution" or "grid spacing". Due to this remark, we have looked for the use of "scale" as we mean it in the article, finding many example that seem to comfort the use of "scale" to actually mean "domain extension", as we understand it:
"Hemispheric-scale modelling of Sulphate and Black Carbon and their Direct Radiative Effects" (1998, Book chapter by Alf Kirkevåg and Øyvind Seland)
"Around the world in 17 days - hemispheric-scale transport of forest fire smoke from Russia in May 2003" (Damoah et al., Atmos. Chem. Phys, 2004)
This, and the very widespread use of "regional scale", "urban scale", not to mean that the smallest eddy resolved by the model has the scale of a urban area, or a region, but that the whole simulation domain has this size, seems to us the most common way to understand "scale" in this context. See, e.g., "Impact of model grid spacing on regional- and urban- scale air quality predictions of organic aerosol.", Stroud et al., 2011, Atmos. Chem. Phys., who clearly use "scale" to specify the domain extension, and another notion, "grid spacing", which is actually equivalent to the smallest resolved eddy - speaking of a chemistry-transport model, whether eddies at this scale are resolved or not depends not on the model itself but on the meteorological data used to force the model. Therefore, as the use of "scale" the way we understand it seems not to be uncommon in our community, and replacing it by "domain size" or an other similar formulation would make many formulations more tedious, we wish to maintain this use of the world scale.

However, to avoid any possible misunderstanding by the readers due to the various possible meaning of the world "scale", we included the following sentence in the Introduction of the paper in order to (hopefully) lift any ambiguity, and also to bring useful information to the readers of this study:
"The typical resolution (grid-spacing) of the simulation domains range from about 4~km for urban-scale domains to about 50km for regional-scale domains (Markakis et al., 2015, Valari et al., 2008)."
(p. 1, l. 49-51)

Pag 2, Introduction State clearly the class of atmospheric-chemistry models which CHIMERE-2016 fits. Is it a coupled online atmospheric-meteorological model? If yes, does it include feedbacks between atmospheric composition and the model dynamics?
Offline model. The following statement has been added :
"CHIMERE-2016 is an offline chemistry-transport model, meaning that it needs to be provided with input meteorological fields, and does not implement any feedback of atmospheric chemistry on atmospheric dynamics." (p. 2, l. 76-80)

How does it compare with other state-of-the-art models like WRF-Chem and COSMO-ART?

As stated above, CHIMERE-2016 is an offline model, so comparison with WRF-CHEM and COSMO-ART is not relevant. This being said, the question of how CHIMERE compares with similar models is

of interest. For that, the reader is now referred to publications from recent intercomparison exercises in which CHIMERE has participated: AQMEII, Eurodelta I and III, CityDelta etc.. This paragraph is in the introduction of the revised manuscript (p. 2, l. 24-49). The interested reader is referred to these publication who detail the characteristics of each of the participating models for many atmospheric trace components.

Pag 3, line 5 State clearly the means of 'CHIMERE core' . Did you say 'dynamic core'?

The notion of "core" for a Chemistry-transport model is actually not clearly defined, and we suppressed it from the manuscript. We initially meant it as the parallelized part of the model (contrary to preprocessors and I/O which was not parallelized), but this is more internal jargon than a recognized notion. In some other parts of the paper, we also removed this ambiguous notion of "core". This statement has been changed as follows (p. 3, l. 11-16):

"Several technical changes were made in the CHIMERE code to improve code scalability: these changes regard the parallelization of many preprocessors into the parallelized section of the model, along with improvement of the parallelization strategy for some parts of the model that were already parallelized in order to improve code scalability."

Page 3, line 20. The expression master/slaves are more common in this context.

OK, we performed this change throughout the paper.

Page 4, lines 1 – 3. In the new version, clarify if the model output is split into several files (each slave writes its model solution in a particular file) or if each slave writes its own sub-domain but in a single file, which comprises the entire domain.

This precision has been added :

Each slave process writes its own sub-domain into a single output netcdf file common to all slaves (p. 3, l. 67-71).

Page 7, Line 20 Include a brief description of the numerical properties of the advection scheme applied in this chimere version to transport scalars.
These precisions have been added a bit earlier than suggested by the Reviewer, because they did not need to be changed for the new model version. The precisions are brought as follows:

In CHIMERE-2017 as in earlier versions, the user can choose between three different options for horizontal transport schemes, namely the basic upwind scheme, the slope-limited Van Leer scheme (Van Leer 1979), and the Piecewise-parabolic method (Collella and Woodward 1984), all of which are examined in the CHIMERE model in lo2009}. Thes three schemes are designed to estimate the trace species concentration at grid cell interfaces in order to convert the mass flux of total air through cell boundaries into mass fluxes for each of the model species through these boundaries. While the implementation of these schemes has needed no change in building the present model version, the estimate of the atmospheric mass flux between neighbouring model grid cells has been revised by switching to a new coordinate system in order to lift model limitations concerning the geographic poles and the date-change lines. These three schemes are designed to be monotonous (because they include the use of slope-limiting algorithms, except the Upwind scheme which does not need the use of such

algorithm), and mass-conservative because of their flux formulation. (from p. 3, l. 87)

Page 8, section 3.1 The new version aims to simulate tracer transport on continental/hemispheric scales with the model top at 200hPa. How the organized vertical transport of pollutants associated with convective and moist plumes are handled in this configuration?

Vertical transport due to deep convection can be activated by the user. If so, the mass fluxes associated to deep convection are estimated using the Tiedke (1989) scheme, and taking into account the fluxes for each model species in the vertical transport and mixing scheme. This is now mentioned in the revised version.

The following sentence has been added :
"Vertical transport on this mesh can be calculated using either a slope-limited Van Leer scheme (Van Leer 1979) or a upwind scheme, depending on user's choice, also taking into account turbulent mixing and, optionally, deep-convection fluxes, following the Tiedke (1989) formulation as described in Menut et al. (2013)."

Page 16, section 5. Describe the numerical solver of the chemical mechanism applied in this model version.

This was not described in the first version of the Manuscript because the solver is the same as in earlier model version. In the revised version of the manuscript, we added the following text :
"As for earlier model versions, the stiff system of partial differential equations resulting from the chemical mechanism is based on the application of a Gauss–Seidel iteration scheme to the 2-step implicit backward differentiation formula, adapted from the algorithm proposed by [Verwer, 1994] More details on this method can be found in Menut (2013)." (p. 6, l. 74-79)

Page 28, line 12 The emission configuration does not agree with the general observed 'umbrela' shape of the volcanic plume. Page 29, Section 7.4

We agree that the volume and distribution of the volcanic emissions is highly uncertain, in terms of emitted mass, vertical profile and deduced optical properties. But the goal of this new experiment was mainly to evaluate the model ability to correctly reproduce long-range transport with the new configuration of the hemispheric model grid.

The simulation outputs discussion lack comparison with observational data. It would be very instructive for the readers to perceive the fidelity of the model transport and AOD simulations.

It is true that comparison to observational data is missing. However, making a comparison with observational data would require a better input regarding volcanic emissions, which we are not able to do at this point. As an example, the parameter $m_{63}$ quantifying the proportion of fine ash / total ash, is highly uncertain, as well as the total ash quantities emitted, etc.

Page 32, Conclusions. Line 16: Is it true that this version 'has the ability to include all types of emissions'

We agree with the Reviewer that this statement is of course exaggerated since some kinds of emissions will always be missing in any model. In the case of CHIMERE, one could mention, for example, the

lightning emission, oceanic DMS emissions, among others. So, we just removed this statement, since the next sentences describe objectively which kinds of emissions have been added since previous model version, which gives more objective and useful information.

---

## Referee Report (RR1)

[referee-annotated manuscript omitted]

---

## Author Response (AR2)

CHIMERE-2017: From urban to hemispheric chemistry-transport modeling
By Mailler, S., et al.
Under review for Geophys. Model Dev.
Answers to Anonymous Reviewer #2 (Minor Revisions)

We would like to thank Anonymous Reviewer No 2 not only for his helpful comments and suggestions but also for providing many suggestions for rewording awkward sentences, which we followed in almost all cases.

Below are the answers to the Reveiwers' comment.

**p. 3, l. 21-26 : Can the model accept input meteorological data with time steps shorter than one hour?**

This is a very good question. Actually, a version with online coupling has already been developed and is now available upon request, as described in the following study :

*Aerosol–radiation interaction modelling using online coupling between the WRF 3.7.1 meteorological model and the CHIMERE 2016 chemistry-transport model, through the OASIS3-MCT coupler*
Briant, R., et al., 2017, Geosci. Model. Dev.

This new online configuration permits to use online coupling for direct radiative forcing, but also, due to the technical upgrades made for the purpose of coupling, permits to force CHIMERE with WRF variables at any prescribes timestep, from the "traditional" hourly timestep downto the models' integration step. However, this model version is still considered a "beta" version ind is available only upon request.

However, with the standard distributed version, meteorological forcing cannot be updated on timesteps shorter than one hour.

The availability of this online-coupled version (so far, only upon request) is now mentioned in the revised version.

**p. 7, l. 47 : Define SNAP here, not down below**

OK, this has been done

**p. 9, l. 41-43 : Awkward sentence, please rephrase**

This sentence was awkward and unclear, we reformulated it using an enumeration of the three existing options

**p. 14, l. 80 : What is meant here by "At that point" ?**

Actually, it just meant that in our view it is useful to bring this precision at that place in the paper. So, the words "at that point" are just useless for the reader, and we

suppressed them.

**p. 15, l. 11-12 : Is it possible to read in other sources of snow cover data?**
Unfortunately, CHIMERE 2017 is not able to use other sources of snow-cover data.

**p. 18, l. 17-19 : Does this underestimation only apply to winter since NO2 is overestimated in summer?**

Yes, this is totally true. Therefore, the statement that a general underestimation of emissions might be the cause for this biases is not presented as sufficient explanation in the revised version of the paper:

"Also, as discussed by Terrenoire (2015), the negative bias could be partly related to the general underestimation of the emissions in the inventory used, especially during the traffic daily peaks. This is in agreement with the relatively high correlation: 0.65 in winter and 0.41 in summer. *However, this would not explain why there is a small positive bias in summer for most stations.*"

**p. 18, l. 57 : What is ammoniac? I assume the authors mean ammonium here.**

Yes, "Ammoniac" has been corrected to "Ammonium"

**p. 19, l. 56 : This is a roughly 20 day spinup, which seems very short for a hemispheric simulation. I suppose it doesn't really matter however since the point of this analysis isn't to compare model to observed concentrations but rather to show the model can transport material across the hemisphere in a reasonable manner.**

Actually, this sentence is a bit misleading since the concentrations do not build up from zero but from LMDZ-INCA climatological values, so that no real spin-up is needed, just an adjustment towards the model's "equilibrium" concentrations. 20 days should obviously enaugh for most species, that are rather short-lived, even though this may not be the case for species such as CO with a longer lifetime.

However, as noted by the Reviewer, this is not critical here since we largely focus on the transport of ash, which largely dominates other species for AOD, and do not focus on other species.

**p. 20, l. 3-5 : It would be nice if the authors could provide several images from this publication here for comparative purposes instead of asking the reader to seek out the article.**

We asked Dr Klüser if he would accept to provide these figures from his *Atmos. Meas. Tech.* Paper for comparison purposes, but his professional adress seems not to work anymore, we then asked to the second author for this permission, but had no answer at the time we have to submit this revised version.

However, since *Atmos. Meas. Tech.* is under CC license, permitting free reuse (with proper attribution), and after consulting the Editorial office of *Geophys. Model Dev.*, we took the liberty to copy these figures as Figs. 10 and 11 of the revised manuscript, changing the text accordingly.

**p. 21, l. 1-5 : Where's the rest of the "either" in this statement? Seems like the authors only stated one part of their thought here.**

Yes, this "either" was not correct, it has been removed and the sentenced has been slightly rephrased without changing its meaning.

**p. 21 : I assume this is supposed to be CHIMERE 2017 and not 2016.**

Yes, this has been changed.

[revised manuscript text omitted]